# DISCRIMINATOR-GUIDED DIFFUSION FOR GENERATING LARGE DIRECTED AND UNDIRECTED GRAPHS

## ABSTRACT

Synthesizing large-scale, realistic (directed) graphs is essential for modeling complex relationships, detecting anomalies, and simulating scenarios where real-world data is sparse, sensitive, or unavailable. While diffusion-based graph generators have shown promising results on small-scale graphs, such as molecular structures, existing models face three key challenges: (1) quadratic time complexity, making them impractical for large graphs; (2) a narrow focus on either structure or node and edge features; and (3) limited exploration of directed graph generation. In this work, we propose **DGDGL**: *Discriminator-Guided Diffusion for Generating Large Directed and Undirected Graphs*. Our approach unifies structure and feature generation for both nodes and edges within a single framework, supporting both directed and undirected graphs. Using graph neural networks and a novel discriminator module, DGDGL guides the denoising process through gradient-based feedback, improving the quality of generated graphs while maintaining linear time complexity with respect to the number of edges. This makes our method scalable to large graphs. We evaluate DGDGL on diverse datasets, including undirected citation networks and directed financial graphs. The results show that our method outperforms existing models with quadratic time complexity. By combining comprehensive support for both directed and undirected graphs, including feature generation for nodes and edges with efficient scalability, DGDGL shows potential for broad use in complex graph-based systems.[1]

## 1 INTRODUCTION

Graphs naturally represent domains in which entities interact in structured and dynamic ways, such as financial networks, social platforms, and communication systems. In many use cases, such as fraud detection using anomaly detection, infrastructure planning by structural analysis, and behavior modeling through scenario simulation, both structural relationships and the features of the nodes and edges are essential. When the data in such real-world tasks are incomplete, sensitive, or costly to obtain, generating large graphs is crucial. To support realistic applications, graph generation should not only focus on topology but also produce representative node and edge features that capture entity attributes and reflect complex interaction patterns.

Graph diffusion-based generative models (GDGMs) encompass two core processes: a forward diffusion process and a reverse denoising process. The forward process injects noise according to a predefined schedule, while the reverse process learns to denoise and reconstruct the original graph. Most existing GDGMs are designed for small undirected graphs such as molecular structures, with relatively low computational cost. This has applications in molecule and protein design, where many alternatives have to be generated. Generating large graphs with GDGMs remains underexplored due to computational challenges. GDGMs have shown strong performance in generating small graphs with both structural and node-level information (Mo et al., 2024; You et al., 2024; Wen et al., 2024; Ayadi et al., 2024). Methods such as DiGress (Vignac et al., 2023) and its generalized variant (Wang et al., 2025) can produce realistic graph topologies along with meaningful node features for graphs containing fewer than 300 nodes. Existing GDGMs face key limitations: the main focus is on

---

[1]The code will be made available upon acceptance.

undirected graphs, they struggle to scale to large graphs, and they often lack the ability to generate informative edge features.

Some GDGMs aim to improve scalability for generating large graphs, but key limitations remain. EDGE (Chen et al., 2023), for example, uses a discrete diffusion process where noise is injected via edge deletion, and 'active' nodes, those whose degrees change, are tracked at each step. Although this improves efficiency, the model cannot fully utilize available information due to insufficient connectivity between nodes. Additionally, its reliance on transformer-based Graph Neural Networks (GNNs) introduces quadratic complexity with respect to the number of active nodes, making it impractical for graphs with thousands of nodes. While SparseDiff (Qin et al., 2023) successfully integrates a sparse graph model and a sparse loss function to enable scalable training, its hard-coded sparsification scheme—typically controlled by a manually chosen edge-fraction hyperparameter—limits the model's expressivity and can compromise generation fidelity. Other models propose alternative strategies. GraphMaker (Li et al., 2024) introduces asynchronous diffusion and decouples structure and feature denoising with mini-batch training and custom message passing, but it assumes rich node features, which are often unavailable in real-world datasets. ARROW-Diff (Bernecker et al., 2025) uses a random-walk-based diffusion process with autoregressive sampling to improve scalability. It lacks formal convergence guarantees and can become unstable in long sequences. While some GDGMs are scalable, they often lack support for edge feature generation and may suffer from memorization or lack convergence guarantees.

To address the lack of scalable graph generative models that jointly handle structure, node features, and edge attributes, we propose **DGDGL**—a discriminator-guided diffusion framework for generating large directed and undirected graphs. DGDGL injects noise into both graph structure and features and learns to denoise them through a GNN-based reverse process. However, a multi-layer message-passing GNN alone is not expressive enough to fully capture the reverse dynamics. To compensate for this limitation, we introduce a discriminator that provides gradient-based guidance, improving sample quality and supported by theory on its role in denoising dynamics. To reduce computational overhead, we inject structural noise through edge deletion. We formulate this process discretely, analyze its impact on GNN-based denoising, and derive the minimum edge density required to preserve connectivity. Our deletion strategy enforces this threshold to ensure reliable reconstruction. Unlike transformer-based models, DGDGL relies on standard GNNs (e.g., GCN (Kipf & Welling, 2017), GIN (Xu et al., 2018), enabling linear scalability with respect to the number of edges. Finally, we demonstrate that the discriminator's output distribution provides a useful signal for downstream tasks such as anomaly detection. Our main contributions are:

- **A Unified framework for directed and undirected graph generation.** We propose a discriminator-guided diffusion framework for graph generation, where a GNN-based discriminator improves structural and feature-level denoising.

- **Theoretical guarantees for structure preservation.** We characterize the impact of edge-removal noise on GNN denoising and derive a lower bound on edge density that ensures reliable reconstruction.

- **Model evaluation.** We evaluate our model on directed and undirected datasets and show strong performance in generating realistic structures alongside node and edge features, with linear computational complexity.

## 2 RELATED WORK

Graph generative models (GGMs) can be broadly categorized into two main classes: statistical models and neural-based models. Statistical models, such as the Erdős–Rényi (ER) (Erdos et al., 1960) and Stochastic Block Model (SBM) (Holland et al., 1983), rely on predefined probabilities to establish edges or structures independently. In contrast, neural models, such as GVAE (Kipf & Welling, 2016) and NetGAN (Bojchevski et al., 2018), aim to learn the underlying data distribution and generate new samples accordingly. Autoregressive models, such as GraphRNN (You et al., 2018), generate graphs sequentially by modeling the adjacency matrix in an autoregressive fashion based on node degrees. However, autoregressive models typically suffer from a computational complexity of at least $O(N^2)$, where $N$ is the number of nodes, limiting their scalability to large graphs.

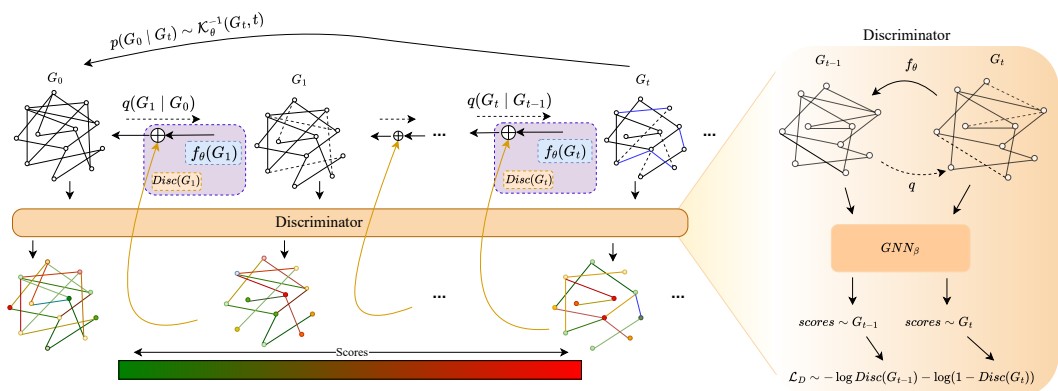

Figure 1: Our generative model includes a forward process that deletes edges and injects noise into node and edge features, starting from the input graph $G_0$. To maintain edge density, random edges (shown in blue) are added. The reverse process uses a GNN to reconstruct $G_0$. A discriminator then assigns realness scores to the generated graphs, visualized from green (realistic) to red (unrealistic).

Recently, Graph diffusion-based generative models have emerged as a promising direction. Existing GDGMs can be categorized into two types: First, Continuous diffusion models (Jo et al., 2022; Niu et al., 2020), which map graph structures into Euclidean space and perform noise injection and denoising in that space. Second, Discrete diffusion models (Xu et al., 2024; Haefeli et al., 2022), which directly inject noise into the graph's discrete structure (e.g., adding or deleting edges). As mentioned earlier, to extend GDGMs to *large-scale graphs*, models such as EDGE (Chen et al., 2023), GraphMaker (Li et al., 2024), and ARROW-Diff (Bernecker et al., 2025) have been proposed.

To the best of our knowledge, no existing diffusion-based model for large graphs supports joint generation of structure and features for both directed and undirected graphs.

Directed graph generation remains largely unexplored. D-VAE (Zhang et al., 2019) proposes a deep generative model for directed acyclic graphs (DAGs) using GNNs with asynchronous message passing to encode computational dependencies more expressively. LayerDAG (Li et al., 2025) introduces an autoregressive diffusion framework for generating realistic DAGs, capturing logical dependencies via bipartite sequence modeling. Table 1 summarizes the capabilities of recent

Table 1: Comparison of graph generation methods by supported graph types and their ability to generate node/edge features. $N$: number of nodes. For NetGAN, $D$: random walk length, $M$: number of sampled walks, $K$: average node degree. For EDGE, $T$: generation steps, $K$: active nodes. For ARROW-Diff, $D$: random walk length, $L$: number of generation steps. For GraphMaker, $F$: number of features per node. Complexity assumes a constant hidden dimension and number of layers.

| Method | Graph Type | | Node Feature | Edge Feature | Complexity |
|---|---|---|---|---|---|
| | Directed | Undirected | | | |
| NetGAN | - | ✓ | - | - | $\mathcal{O}(MD + NK)$ |
| VGAE | - | ✓ | ✓ | - | $\mathcal{O}(N^2)$ |
| GraphRNN | - | ✓ | - | - | $\mathcal{O}(N^2)$ |
| EDGE | - | ✓ | - | - | $\mathcal{O}(T\max(|E|, K^2))$ |
| GraphMaker | - | ✓ | ✓ | - | $\mathcal{O}(FN)$ |
| ARROW-Diff | - | ✓ | - | - | $\mathcal{O}(L(ND + |E|))$ |
| D-VAE | ✓ | - | ✓ | - | $\mathcal{O}(N^2)$ |
| LayerDAG | ✓ | - | - | - | $\mathcal{O}(N^2)$ |
| DGDGL (ours) | ✓ | ✓ | ✓ | ✓ | $\mathcal{O}(T|E|)$ |

GGMs, highlighting our model's ability to generate both node and edge features with linear complexity. Model complexities are cited from the original papers.

## 3 PROPOSED METHOD

We propose **DGDGL**, a discriminator-guided diffusion model for generating large directed and undirected graphs. Our approach builds on diffusion models, which iteratively refine noisy graph samples by learning the joint distributions of node features ($p(\mathbf{X})$), edge structures ($p(\mathbf{E})$), and edge features ($p(\mathbf{F})$). By modeling these components, we approximate the true graph distribution ($p(G)$), enabling the generation of diverse and realistic graph samples. To further enhance sample quality, a GNN-based discriminator—inspired by (Kim et al.; Wang et al.)—guides the denoising process by providing informative feedback during training. We begin by describing the forward diffusion pro-

cess, where noise is injected into both graph structure and features, followed by the reverse process learned through denoising. Next, we introduce the discriminator's integration into the framework, provide theoretical justification for its role in guiding the reverse process, and present its application to anomaly detection. Fig. 1 illustrates the overall workflow.

## 3.1 PRELIMINARIES

We represent a graph as $G = (V, E, \mathbf{X}, \mathbf{F})$, where $V$ and $E$ denote the sets of nodes and edges. Each node $v_i \in V$ has a feature vector $\mathbf{X}_i \in \mathbb{R}^{d_v}$, and each edge $e_{ij} \in E$ has a feature vector $\mathbf{F}_{ij} \in \mathbb{R}^{d_e}$ leading to respectively node feature matrix $\mathbf{X} \in \mathbb{R}^{|V| \times d_v}$ and edge feature matrix $\mathbf{F} \in \mathbb{R}^{|E| \times d_e}$.

**Graph diffusion Process.** GDGMs define a forward process $q(G_t \mid G_{t-1})$ for $t = 0, \ldots, T$ that gradually corrupts a graph $G_0 \sim (p(\mathbf{X}), p(\mathbf{E}), p(\mathbf{F}))$ into noise $G_T \sim \mathcal{N}(0, I)$, and a reverse process $p(G_{t-1} \mid G_t)$ learned via a denoising network.

A message passing graph neural network (GNN) parameterizes the reverse process as follows:

$$\mathbf{h}_v^{(l+1)} = \phi\left(\mathbf{h}_v^{(l)}, \square_{u \in N(v)} \psi(\mathbf{h}_v^{(l)}, \mathbf{h}_u^{(l)}, \mathbf{e}_{uv}^{(l)})\right), \tag{1}$$

$$\mathbf{h}_v^{(0)} = \mathbf{X}_v, \mathbf{e}_{uv}^{(0)} = \mathbf{F}_{uv},$$

where $h_v^{(l)}$ is the hidden state of node $v$ at time step $t$ in layer $l$, $N(v)$ is its neighborhood, $\phi$ and $\psi$ are learnable functions, and $\square$ denotes aggregation. In a directed graph, $N(v)$ denotes the set of in-neighbors. After $L$ layers, the information at node $v$ comes from its $L$-hop neighborhood, defined as $B_L(v) = \{u \in V \mid \text{dist}(u, v) \leq L\}$, where $\text{dist}(u, v)$ is the length of the shortest path from $u$ to $v$.

## 3.2 GRAPH DIFFUSION MODEL SETUP

Let $G_0 = (V, E_0, \mathbf{X}_0, \mathbf{F}_0)$ be the original graph, where $\mathbf{X}_0$ are the node features and $\mathbf{F}_0$ are the edge features. At diffusion step $t$, $G_t = (V, E_t, \mathbf{X}_t, \mathbf{F}_t)$ evolves through edge and noise injection. Since edge addition introduces significant computational overhead for large graphs, we apply structural noise through edge deletion rather than addition:

**Edge Deletion.** The forward process applies a stochastic deletion operator $\mathcal{D}_t$ at each step $t$,

$$E_{t+1} = \mathcal{D}_t(E_t, \epsilon_t),$$

where each edge existing at step $t$ is independently removed with probability $\epsilon_t$. For any edge $(u, v) \in E_t$,

$$q_e(e_{uv}^{(t+1)} = 0 \mid e_{uv}^{(t)} = 1) = \epsilon_t, \qquad q_e(e_{uv}^{(t+1)} = 1 \mid e_{uv}^{(t)} = 1) = 1 - \epsilon_t,$$

and non-edges remain unchanged ($q_e(e_{uv}^{(t+1)} = 0 \mid e_{uv}^{(t)} = 0) = 1$). More details are provided in Appendix B.

**Feature Noise Injection.** For node features $\mathbf{X}$ and edge features $\mathbf{F}$, we use the standard variance-preserving diffusion parameterization: $\mathbf{X}_t = \sqrt{\alpha_t} \mathbf{X}_0 + \sqrt{1 - \alpha_t} \eta_{\mathbf{X}_t}$ and $\mathbf{F}_t = \sqrt{\alpha_t} \mathbf{F}_0 + \sqrt{1 - \alpha_t} \eta_{\mathbf{F}_t}$, where $\eta_{\mathbf{X}_t}, \eta_{\mathbf{F}_t} \sim \mathcal{N}(0, I)$. Here, $\alpha_t = \prod_{s=1}^{t}(1 - \beta_s)$ is the cumulative product of the noise schedule, and $\beta_t \in (0, 1)$ controls the injected noise at each step.

Let $f_\theta$, a GNN parameterized by $\theta$, serve as the core of the reverse transition kernel, $\mathcal{K}_\theta^{-1}(G_t, t)$. It is trained to reconstruct the original graph $G_0$ from a random graph $G_T$. We approximate $G_{t-1}$ from $G_t$ using $p(G_{t-1} \mid G_t) \approx \mathcal{K}_\theta^{-1}(G_t, t)$. Fig. A1 illustrates the architecture of $f_\theta$, where, formally, at each time step $t$, we have:

$$\left[\hat{\mathbf{X}}_t, \hat{\mathbf{F}}_t, \mathbf{s}_t, \hat{\mathbf{e}}_t\right] = f_\theta(G_t), \tag{2}$$

where $\hat{\mathbf{X}}_t, \hat{\mathbf{F}}_t, \mathbf{s}_t \in \mathbb{R}^{|E_t|}, \hat{\mathbf{e}}_t \in \mathbb{R}^{|E_t|}$ are the predictions of the node and edge features, the edge scores, and the noise predictions of the edge score, respectively. We apply multi-head edge aware-ness to capture structural differences between time steps $t$ and $t - 1$. For more details, please refer

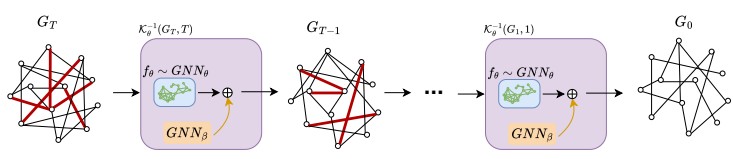

Algorithm 1: Generation process

1: **Input:** $f_\theta, T, \rho_T^{\min}, \alpha$
2: Sample $E_T \sim \text{Bernoulli}(\rho_T^{\min})$; $\mathbf{X}_T \sim \mathcal{N}(0, I)$; $\mathbf{F}_T \sim \mathcal{N}(0, I)$
3: $G_T \leftarrow (E_T, \mathbf{X}_T, \mathbf{F}_T)$
4: **for** $t = T$ to 1 **do**
5:     $\hat{\mathbf{X}}_t, \hat{\mathbf{F}}_t, \mathbf{s}_t, \hat{\mathbf{e}}_t = f_\theta(G_t)$
6:     $E_{t-1} \leftarrow \hat{E}_t$       ▷ Eq. 4
7:     $\mathbf{X}_{t-1} \leftarrow \hat{\mathbf{X}}_t$       ▷ Eq. 9
8:     $\mathbf{F}_{t-1} \leftarrow \hat{\mathbf{F}}_t$       ▷ Eq. 10
9: **end for**
10: **return** $G_0$

Figure 2: Graph generation starts from a random graph. The model $f_\theta$ iteratively refines noisy edges using node and edge features to produce a realistic structure. Node and edge features are not depicted.

to Appendix A.1. To align the model with our goal of generating features and structure jointly, we optimize $f_\theta$ using the following loss function:

$$\mathcal{L}_{\text{recon}} = \mathbb{E}_{G_0,\, t\sim\mathcal{U}(0,T)} \left[ \left\| \hat{\mathbf{X}}_t - \mathbf{X}_0 \right\|^2 + \left\| \hat{\mathbf{F}}_t - \mathbf{F}_0 \right\|^2 + \| \hat{\mathbf{e}}_t - \mathbf{e}_0 \|^2 + \| \mathbf{s}_t - \mathbf{e}_0 \|^2 \right]. \quad (3)$$

To enable accurate reconstruction during generative diffusion, we need to access a minimum amount of information about the structure. To achieve this, we introduce the minimum edge density $\rho_t = \frac{|E_t|}{|E_0|}$ at each time step $t$, which is required to preserve sufficient structure for effective GNN operation. We introduce a threshold $\Omega$, requiring that each node $v$ has at least $\Omega$ nodes in its $L$-hop neighborhood, i.e., for each $v$: $|B_L^{(t)}(v)| \geq \Omega$. This constraint informs the control of $\rho_t$ during the diffusion process, as insufficient density hinders graph recovery by the GNN, motivating the following corollary:

**Corollary 3.1** *A GNN with $L$ layers can recover $G_{t-1}$ from $G_t$ if $\rho_t^{\min} \geq \left( \frac{\Omega}{d_{org}^L} \right)^{1/L}$, where $d_{org}$ denotes the average degree for undirected graphs or average in-degree for directed graphs.*

On the other hand, edge deletion disrupts effective paths between nodes, potentially causing GNNs to suffer from over-squashing and degrading the denoising process. Over-squashing occurs when information from distant nodes is compressed into fixed-size representations, leading to the loss of critical signals in graphs with bottlenecks or sparse connectivity (Black et al., 2023). This happens when the receptive field grows faster than the model's ability to retain meaningful information. To clarify the link between over-squashing and reduced connectivity due to edge deletion, we state the following:

**Proposition 3.1** *Edge deletion, by reducing paths between nodes, raises over-squashing, which prevents GNNs from propagating structural information.*

Mathematical derivations of Corollary 3.1 and Proposition 3.1 are provided in Appendix C and Appendix D, respectively. According to the above, our edge deletion process is designed to satisfy $|\tilde{E}_t| \geq \rho_t^{\min} \cdot |E_0|$. If the graph at time step $t$ becomes sparser than this threshold, we randomly and independently add edges -sampled from a uniform distribution- until the desired edge density is restored. Therefore, the edge transition is modified as $\mathcal{D}_t(E_0, \eta_{\epsilon_t})$, where the cumulative removal probability is constrained to ensure minimum edge density: $\eta_{\epsilon_t} = \min\left\{ 1 - \prod_{i=1}^{t}(1 - \epsilon_i),\, 1 - \rho_t^{\min} \right\}$.

**Generation (sampling) Process** The reverse process starts from a random graph $G_T = (V, E_T, \mathbf{X}_T, \mathbf{F}_T)$, where $E_T$ contains a set of randomly sampled edges ($E_T \sim \text{Bernoulli}(\rho_T^{\min})$), and node and edge features are initialized as $\mathbf{X}_T \sim \mathcal{N}(0, I)$ and $\mathbf{F}_T \sim \mathcal{N}(0, I)$, respectively. Edges are then progressively revised based on predicted noise and edge scores derived from these initial features. Fig. 2 and Algorithm 1 illustrate the generation process.

Edge Update: At step $t$, edges are removed based on the edge scores and predicted noise. The set of edges $\hat{E}_t$ is updated as:

$$\hat{E}_t = \left\{ (u, v) \in E_T \,\middle|\, s_{uv,t} > \hat{e}_{uv,t} \right\} \quad (4)$$

Feature Reconstruction: Node and edge features are reconstructed using the predicted noise at time step $t$:

$$\mathbf{X}_{t-1} = \frac{\mathbf{X}_t - \sqrt{1 - \alpha_t}\hat{\mathbf{X}}_t}{\sqrt{\alpha_t}} + \sqrt{1 - \alpha_t}\eta_{\mathbf{X}_t}, \tag{5}$$

$$\mathbf{F}_{t-1} = \frac{\mathbf{F}_t - \sqrt{1 - \alpha_t}\hat{\mathbf{F}}_t}{\sqrt{\alpha_t}} + \sqrt{1 - \alpha_t}\eta_{\mathbf{F}_t}. \tag{6}$$

**Reverse Transition Kernel**  The reverse transition kernel models the reconstruction of edges and features in a unified process:

$$p(G_{t-1} \mid G_t) = p(\mathbf{X}_{t-1}, \mathbf{F}_{t-1}, E_{t-1} \mid \mathbf{X}_t, \mathbf{F}_t, E_t) \tag{7}$$
$$\approx \mathcal{K}_\theta^{-1}(\mathbf{X}_t, \mathbf{F}_t, E_t, t) = \mathcal{K}_\theta^{-1}(G_t, t)$$

Here, $\mathcal{K}_\theta^{-1}$ represents the combined edge update and feature denoising operations. The output of $\mathcal{K}_\theta^{-1}$ forms a sequence of transition graphs for $t = T, \ldots, 1$, with $t = 0$ producing the final generated graph.

### 3.3  DISCRIMINATOR

As indicated above, our model aims to learn a mapping from the noise distribution $p_{G_T}$ to the data distribution $p_{G_0}$. However, when using a simple backbone for the sake of scalability, this mapping alone is not sufficiently expressive to capture the full complexity of the reverse process. As we demonstrate in the ablation study, additional guidance is necessary to ensure high-quality generation. Therefore, we incorporate a discriminator to guide and improve the mapping by distinguishing between realistic and unrealistic graphs. Specifically, we enforce the discriminator to treat the graph at time step $t + 1$ as less realistic than the graph at time step $t$ in the diffusion process. To distinguish real from generated graphs, we employ a discriminator $\mathbf{S_H}, \mathbf{S_F}, \mathbf{s_e} = Disc(G)$, based on a GNN, where $\mathbf{S_H} \in \mathbb{R}^{|V| \times d_v}$, $\mathbf{S_F} \in \mathbb{R}^{|E| \times d_e}$ and $\mathbf{s_e} \in \mathbb{R}^{|E|}$ are feature nodes, features edges, and edge scores, respectively. The discriminator, $\text{GNN}_\beta$, aims to assign low scores to real node features or edges and high scores to generated (less realistic) ones, providing feedback to guide the graph generation process. Fig. A1 shows the schematic of $Disc$; further details are provided in Appendix A.2.

Let $G_t = \mathcal{K}_\theta^{-1}(G_{t+1}, t + 1)$ denote the graph generated from $G_{t+1}$ using the reverse process $f_\theta$. We then define the following objective:

$$\mathcal{L}_D = -\mathbb{E}_{G_t \sim p_{G_0}}\left[\log Disc(G_t)\right] - \mathbb{E}_{G_{t+1} \sim p_{G_T}}\left[\log(1 - Disc(G_{t+1}))\right]. \tag{8}$$

**Discriminator guidance**  The $Disc$ incorporates graph reconstruction within the reverse process as follows:

$$\mathbf{X}_{t-1} = \frac{\mathbf{X}_t - \sqrt{1 - \alpha_t}\hat{\mathbf{X}}_t}{\sqrt{\alpha_t}} + \sqrt{1 - \alpha_t}\eta_{\mathbf{X}_t} + \lambda \cdot \nabla_{\mathbf{X}_t} \log(\mathbf{S_{H}}_t), \tag{9}$$

$$\mathbf{F}_{t-1} = \frac{\mathbf{F}_t - \sqrt{1 - \alpha_t}\hat{\mathbf{F}}_t}{\sqrt{\alpha_t}} + \sqrt{1 - \alpha_t}\eta_{\mathbf{F}_t} + \lambda \cdot \nabla_{\mathbf{F}_t} \log(\mathbf{S_{F}}_t), \tag{10}$$

where $\lambda$ is a hyperparameter controlling the strength of the discriminator guidance. We claim that the guidance provided by $Disc$, improves the learned mapping from noise to the data distribution. Accordingly, we state the following:

**Proposition 3.2** *Let $p_{G_T}$ denote the initial noise distribution in a graph diffusion model, and let $p_{G_0}$ and $p_\theta$ represent the original data distribution and the learned transformation distribution via $f_\theta$, respectively. Then, the discriminator-guided update (Eq. 9 and Eq. 10) approximates the Wasserstein gradient flow of the Kullback-Leibler divergence $\mathrm{KL}(p_\theta(G) \,\|\, p_{G_0})$, thus improving the transport from $p_{G_T}$ to $p_{G_0}$.*

A GNN's capacity to distinguish real and generated graphs is closely tied to its ability to model global structural properties. We show that undirected graphs require at least $\frac{D}{2}$ layers, while directed graphs require at least $D_{\text{dir}}$, where $D$ and $D_{\text{dir}}$ are their respective diameters. Given that in general the GNN expressivity power is equivalent to the WL test (Xu et al., 2018), we can state the following:

**Corollary 3.2** *If two graphs require $L_L$ iterations of the WL test to be distinguished, a GNN must also have at least $L_L$ layers to reach the same level of expressiveness.*

Corollary 3.2 allows us to have the following statements:

**Corollary 3.3** *A GNN can fully distinguish real from unrealistic graphs only if the number of layers $L$ satisfies $L \geq \frac{D}{2}$ for **undirected** graphs and $L \geq D_{dir}$ for **directed** graphs.*

Therefore, these considerations should be reflected in the design of $Disc$. A detailed discussion of Proposition 3.2, Corollaries 3.2, and 3.3 is provided in Appendix E, F, and G, respectively. Since we apply a GNN-based model for denoising, our approach maintains linear computational complexity with respect to the number of edges at each diffusion step. A detailed discussion is provided in Appendix A.3.

### 3.4 DISCRIMINATOR APPLICATION

Analyzing anomalies through data distributions makes generative models well-suited for this task. By leveraging the self-supervised objective of $Disc$, we identify deviations from the learned normal distribution, enabling effective anomaly detection as a downstream application. An anomaly refers to a node, edge, or subgraph that lies in the low-density region of the node feature distribution $p(\mathbf{X})$, edge structure $p(E)$, or edge feature distribution $p(\mathbf{F})$, thereby deviating from a typical graph (Akoglu et al., 2015; Ruff et al., 2021). The generative model $f_\theta$, trained via the variational bound, is optimized over samples from $p_{G_0}$ and thus learns to reconstruct graphs close to the data manifold of normal attributes. The discriminator, $Disc$, is trained to distinguish denoised graphs $G_t \sim p_{G_0}$ from noisy ones $G_{t+1} \sim p_{G_T}$. Under this setting, the Bayes-optimal discriminator is: $\text{Disc}^*(G) = \frac{p_{G_T}(G)}{p_{G_0}(G) + p_{G_T}(G)}$.

Let an input graph $G^{(a)}$ contain anomalous nodes or edges. Since $G^{(a)} \not\sim p_{G_0}$ (attributes off-manifold) and $G^{(a)} \approx$ a noisy variant (i.e., structurally similar to samples from $p_{G_T}$), we have: $Disc(G^{(a)}) = \frac{p_{G_T}(G^{(a)})}{p_{G_0}(G^{(a)}) + p_{G_T}(G^{(a)})} \approx 1$, since $p_{G_0}(G^{(a)}) \approx 0$ while $p_{G_T}(G^{(a)}) > 0$ due to its noise-like appearance. Therefore, $\mathbf{S_H}$ and $\mathbf{s}_e$ capture deviations in node and edge inputs, which we rank as anomaly scores.

## 4 EXPERIMENTS

Since our proposed model includes both a generator and a discriminator, we evaluate it along two main tracks: generation, and anomaly detection as a downstream task related to the discriminator.

Table 2: Dataset statistics.

| Dataset | Nodes | Edges | Node Feature | Edge Feature | Type |
|---|---|---|---|---|---|
| Bitcoin-Alpha | 3783 | 24186 | - | 3 | directed |
| Bitcoin-OTC | 5881 | 35592 | - | 3 | directed |
| Elliptic | 18945 | 21660 | 186 | 3 | directed |
| Cora | 2708 | 10556 | 1433 | - | undirected |
| Citeseer | 3327 | 9104 | 3703 | - | undirected |
| Pubmed | 19717 | 88648 | 500 | - | undirected |

### 4.1 DATASETS AND METRICS

To evaluate our model, we consider a diverse range of graph datasets, from citation networks (Sen et al., 2008) to financial data (Kumar et al., 2016; 2018; Elmougy & Liu, 2023b). Table 2 shows dataset statistics; see Appendix I for more details. We consider the Maximum Mean Discrepancy (MMD) (Gretton et al., 2012) between the input and generated graphs in terms of node degree distributions, clustering coefficient distributions, and node/edge feature distributions. For the MMD metric, smaller is better. We also compare the statistics of the generated graphs with those of the corresponding input graphs in terms of the power law exponent of the degree (PLE), characteristic path length (CPL) (Chanpuriya et al., 2021), and the average, maximum, and variance of node degrees. We additionally report the edge overlap (EO) between the generated and original graphs,

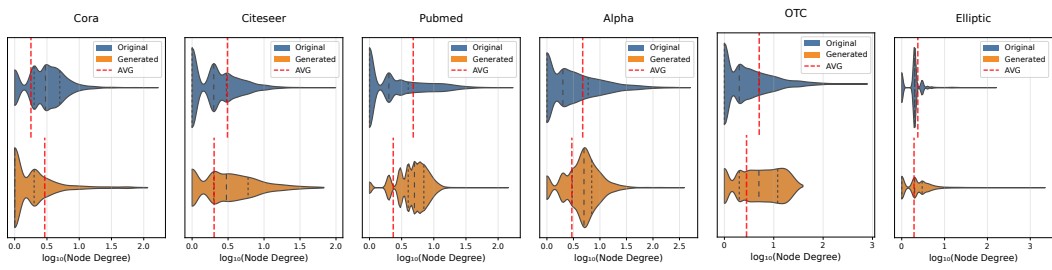

Figure 3: Degree distribution comparison across six datasets (plot width indicates node density)

as a high EO may indicate potential memorization (Chen et al., 2023). For implementation details, please refer to Appendix A.4[2].

Table 3: MMD scores for degree distribution, clustering coefficient, node features, and edge features across three datasets. A '-' indicates that the model is either incapable of generating that feature or the dataset does not contain it. Lower scores are better; best and second-best results are highlighted in **bold** and underlined, respectively.

| | MMD ($\times 10^{-2}$) ↓ | | | | | | | | | | | |
|---|---|---|---|---|---|---|---|---|---|---|---|---|
| | **Citeseer** | | | | **OTC** | | | | **Elliptic** | | | |
| | Deg. | Clus. | NodeF | EdgeF | Deg. | Clus. | NodeF | EdgeF | Deg. | Clus. | NodeF | EdgeF |
| GraphRNN | **2.01**±0.14 | 1.50±0.11 | - | - | 9.95±0.33 | 7.81±0.28 | - | - | **1.19**±0.09 | 1.73±0.12 | - | - |
| GraphMaker | 17.50±0.70 | 23.50±1.10 | **0.17**±0.01 | - | 53.10±1.20 | 34.70±0.90 | - | - | 10.10±0.65 | 10.80±0.42 | 1.18±0.09 | - |
| EDGE | 4.78±0.19 | 1.33±0.07 | - | - | 5.31±0.21 | 3.11±0.12 | - | - | 6.19±0.25 | 2.07±0.10 | - | - |
| ARROW-Diff | 4.13±0.18 | **1.12**±0.05 | - | - | 5.23±0.19 | 2.77±0.11 | - | - | 6.17±0.22 | 1.01±0.06 | - | - |
| DGDGL (ours) | 3.97±0.16 | 1.32±0.06 | 0.21±0.02 | - | **4.67**±0.17 | **2.01**±0.09 | - | **5.50**±0.18 | 6.28±0.23 | **0.97**±0.05 | **0.14**±0.01 | **5.78**±0.19 |

## 4.2 RESULTS

After generating 32 samples for each graph, we compute the evaluation metrics. Table 3 reports the average MMD results across models, indicating the distance between the distributions of generated graphs and their corresponding input graphs in terms of degree, clustering coefficient, and node or edge feature distributions.

As shown, models like GraphMaker, which rely heavily on node features, perform better in feature generation for datasets such as Citeseer, which contain rich node features. In Table 3, Deg and Clus evaluate the structural generation capabilities of the models. As the number of nodes increases, as in Elliptic, or when node and edge features are less informative (e.g., Alpha and OTC), our model demonstrates stronger performance. As shown in Table 3, none of the baseline models for large graph generation support both edge and node features or handle both directed and undirected graphs, whereas our proposed model supports all of these capabilities.

Beyond distribution-level evaluation, Table 4 presents statistics comparing a randomly selected example from the 32 generated graphs with its original input counterpart. These statistics include

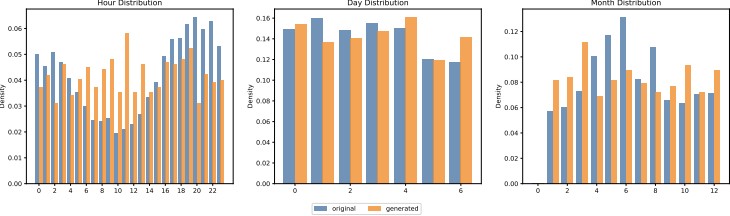

Figure 4: Generated vs. original edge features on OTC.

the average ($AVG_d$), maximum ($MAX_d$), and variance ($VAR_d$) of node degrees, as well as PLE and CPL. The results in Table 4 highlight important trade-offs in graph structure. A high $MAX_d$ is only meaningful when considered alongside the $AVG_d$; for example, high $MAX_d$ with low $AVG_d$ often indicates greater structural diversity and higher $VAR_d$. This variance reflects the model's ability to capture diverse graph topologies. EO helps assess whether a model generalizes or simply memorizes the input. While models like ARROW-Diff perform reasonably well, their high EO suggests limited generalization due to potential overfitting, often resulting in low $VAR_d$.

---

[2]For the baselines, we select efficient yet comparable models that are capable of handling large-graph generation, and we do not consider results that lead to out-of-memory issues.

Table 4: Statistics of the generated graphs compared to their input counterparts. For PLE and CPL, values closer to the ground truth are better. The EO value for the ground truth represents its self-overlap. The best and second-best results are highlighted in **bold** and underlined, respectively.

| | Cora | | | | | | Pubmed | | | | | | Alpha | | | | | |
|---|---|---|---|---|---|---|---|---|---|---|---|---|---|---|---|---|---|---|
| | $EO\downarrow$ | PLE | CPL | $AVG_d\downarrow$ | $MAX_d\uparrow$ | $VAR_d\uparrow$ | $EO\downarrow$ | PLE | CPL | $AVG_d\downarrow$ | $MAX_d\uparrow$ | $VAR_d\uparrow$ | $EO\downarrow$ | PLE | CPL | $AVG_d\downarrow$ | $MAX_d\uparrow$ | $VAR_d\uparrow$ |
| ground truth | 100 | 1.83 | 6.31 | 3.89 | 168 | 27.3 | 100 | 2.21 | 6.33 | 4.49 | 171 | 55.2 | 100 | 1.24 | 3.57 | 7.46 | 510 | 401.95 |
| GraphRNN | 0.01 | 1.9 | **5.39** | **1.79** | 20 | 2.31 | 0.1 | 1.78 | 4.3 | **5.1** | 15 | 8.6 | 0.1 | 0.88 | 1.71 | **3.7** | 7.1 | 8.1 |
| GraphMaker | 3.7 | 1.3 | 2.64 | 8.8 | 25 | 17.8 | 10.7 | 1.3 | 2.74 | 12.4 | 30 | 17.6 | 9.2 | 0.35 | 1.37 | 15.6 | 70 | 23.4 |
| EDGE | 2.1 | 1.76 | 4.98 | 7.63 | 87 | 21.8 | 8.2 | 2.15 | **6.23** | 8.3 | 91 | 36.8 | 3.7 | 0.91 | 2.1 | 11.3 | 283 | 57.8 |
| ARROW-Diff | 41.2 | 1.66 | 4.82 | 5.73 | 84 | 13.7 | 51.3 | 1.89 | 5.27 | 6.01 | **147** | 39.3 | 67.8 | 0.99 | 2.05 | 8.1 | 113 | 23.6 |
| DGDGL (ours) | 0.0 | **1.82** | 4.84 | 6.38 | **93** | **35.6** | 0.0 | **2.19** | 5.58 | 5.7 | 145 | **47.4** | 0.0 | **1.29** | **3.65** | 7.37 | **300** | **123.4** |

Table 5: Performance metrics at various diffusion time steps for Citeseer and OTC datasets.

| | Citeseer | | | | | | | | OTC | | | | | |
|---|---|---|---|---|---|---|---|---|---|---|---|---|---|---|
| | T | $EO\downarrow$ | PLE | CPL | $AVG_d\downarrow$ | $MAX_d\uparrow$ | $VAR_d\uparrow$ | | T | $EO\downarrow$ | PLE | CPL | $AVG_d\downarrow$ | $MAX_d\uparrow$ | $VAR_d\uparrow$ |
| ground truth | | 100 | 1.88 | 9.32 | 2.73 | 99 | 11.4 | ground truth | | 100 | 1.24 | 3.57 | 7.3 | 795 | 530.7 |
| | 64 | 9.4 | 1.78 | 3.52 | 5.1 | 49 | 10.2 | | 64 | 1.9 | 1.15 | 2.3 | 8.5 | 84 | 33.4 |
| | 128 | 1.3 | 1.77 | 3.51 | 5.5 | 57 | 15.2 | | 128 | 0.7 | 1.19 | 3.01 | 8.9 | 97 | 61.2 |
| | 256 | 0.0 | 1.90 | 5.52 | 4.03 | 71 | 51.4 | | 256 | 0.0 | 1.39 | 3.62 | 9.1 | 377 | 131.5 |
| | 512 | 0.9 | 1.80 | 2.56 | 5.8 | 75 | 40.7 | | 512 | 0.2 | 1.31 | 3.1 | 10.1 | 387 | 107.2 |
| | 1024 | 2.1 | 1.79 | 2.78 | 5.6 | 77 | 41.4 | | 1024 | 1.9 | 1.21 | 3.2 | 9.5 | 370 | 128.1 |

When evaluating degree statistics jointly— $VAR_d$, $MAX_d$, and $AVG_d$—our model shows greater flexibility and diversity in generated structures, as illustrated in Fig. 3. PLE further reflects how well models preserve the scale-free properties of real graphs. Table 4 shows our model closely matches the input PLE distribution and achieves the best PLE across all datasets, demonstrating effective distribution learning through our GNN-based architecture. For directed datasets like Alpha and OTC, distributional and statistical evaluations are performed based on node in-degree. Fig. 4 shows the generated edge attributes, where timestamps are decomposed into month, day, and hour. As illustrated, our model generates edge attributes that closely match the original features. Detailed results, including those for anomaly detection, are provided in Appendix H.

**Anomaly detection** To evaluate the discriminator's effectiveness in self-supervised anomaly detection, we focus on the Alpha and OTC datasets (Fig. 5), where edge labels range from –10 to +10. Following prior work (Elmougy & Liu, 2023a; Li et al., 2023), edges with negative labels are treated as anomalies. As shown in Fig. 6, $Disc$ assigns higher anomaly scores to these negative-labeled edges, indicating its ability to detect anomalous patterns without supervision. We also compare $Disc$ with a baseline GNN model, GNN$_{base}$, which shares the same architecture but is trained from scratch for edge regression using the loss $\mathcal{L}_{D_{base}} = \left\| \hat{s}_e^{base} - e_0 \right\|^2$. Fig. 6 shows both $\hat{s}_e^{base}$ and $s_e$, where $s_e$ (produced by $Disc$) exhibits a stronger correlation with the negatively labeled edges, indicating its superior anomaly detection performance.

**Time diffusion steps impact.** Here, we investigate the model's performance under different settings. We explore the effect of the number of diffusion steps to understand how time steps influence the generation process. Table 5 presents part of our results, confirming that increasing the number of diffusion steps does not necessarily lead to better performance. In general, more diffusion steps tend to result in a higher maximum degree, often desirable, accompanied by a reasonably low average degree. Based on our observations, there is no clear or consistent relationship between the number of diffusion steps (beyond 256) and generation performance. More detailed ablation studies, including the effect of $Disc$ on generation and the satisfaction of $\rho^{min}$, are provided in Appendices H.1 and H.2, respectively.

Figure 5: Edge label.

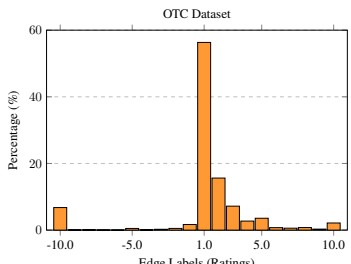

Figure 6: Anomaly scores.

## 5   CONCLUSION

We present a scalable diffusion model for generating large graphs with both structural and feature information. Our approach uses graph neural networks to efficiently generate directed and undirected graphs along with their node and edge features, achieving linear complexity that scales with graph size. The model includes a discriminator to improve generation quality, which can also be applied as a downstream example task. Experiments across multiple datasets show our method excels at graph generation, and demonstrates competitive performance in anomaly detection compared to dedicated detection models.

**Limitations.** While the model effectively captures both structural and attribute distributions, it tends to generate graphs with higher average degrees. This results from GNN-based message passing, which emphasizes the local neighborhood structure, whereas global graph properties are also important. The current proposed approach leverages only in-degree, which in practice is more informative and crucial for directed graphs (Page et al., 1999), while leaving out-degree patterns unaddressed. These limitations point to future directions, such as balancing degree distributions and extending the framework to support out-degree modeling.

**Ethics Statement.** This work presents a diffusion-based method for generating large graphs, using only public benchmark datasets and no human or sensitive data[3].

---

[3]AI (LLMs) were used exclusively for writing polishing.

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

# Appendix

## Contents

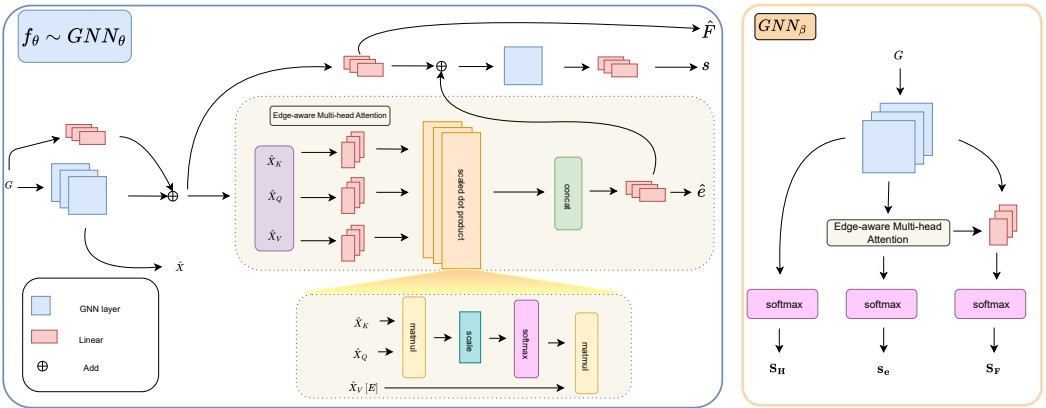

Figure A1: Denoising model, $f_\theta$, with edge-aware attention (left) and discriminator, $Disc$, structure (right).

# A ARCHITECTURE AND IMPLEMENTATION DETAILS

Here, we provide a detailed explanation of the operations of $f_\theta$ and $Disc$.

## A.1 $f_\theta$ ARCHITECTURE

At diffusion step $t$, let $G_t = (V, E_t, \mathbf{X}_t, \mathbf{F}_t)$. The node representations are updated as:

$$\mathbf{h}_v^{(l+1,t)} = \phi\left(\mathbf{h}_v^{(l,t)}, \square_{u \in N(v)} \psi\left(\mathbf{h}_v^{(l,t)}, \mathbf{h}_u^{(l,t)}, \mathbf{e}_{uv}^{(l,t)}\right)\right), \tag{A1}$$

where $\phi$ and $\psi$ are learnable functions, and $\square$ denotes a permutation-invariant aggregation operator. After $L$ message-passing layers, the final node representations are stacked as $\mathbf{H}_t$, and the node feature matrix is updated as $\hat{\mathbf{X}}_t = \mathbf{H}_t^{(L)}$. The edge features are then obtained as follows:

$$\hat{\mathbf{f}}_{i \to j,t} = \text{MLP}(\hat{\mathbf{X}}_t[i] \,\|\, \hat{\mathbf{X}}_t[j]), \quad \forall (i,j) \in E_t \tag{A2}$$

$$\hat{\mathbf{F}}_t = stack(\hat{\mathbf{f}}_{i \to j,t}) \tag{A3}$$

where $\|$ denotes the concatenation operation and $\hat{\mathbf{F}}_t$ is the whole matrix of edge features at time step $t$.

**Edge-Level Multi-Head Attention**  Inspired by graph attention models (Velickovic et al., 2018; Vaswani et al., 2017), and given the node feature matrix $\mathbf{X}_t \in \mathbb{R}^{n \times d}$ and a set of edges $E_t \subseteq V \times V$, we define edge-wise attention using multi-head QKV attention.

For each head $k = 1, \ldots, K$ and each edge $(i,j) \in E_t$, compute:

$$\mathbf{q}_i^{(k)} = \mathbf{W}_Q^{(k)} \mathbf{x}_i, \quad \mathbf{k}_j^{(k)} = \mathbf{W}_K^{(k)} \mathbf{x}_j, \quad \mathbf{v}_{ij}^{(k)} = \mathbf{W}_V^{(k)} \cdot \gamma(\mathbf{x}_j, \mathbf{e}_{ij}), \tag{A4}$$

where $\gamma(\cdot)$ combines node and edge features (e.g., via concatenation or MLP), and $\mathbf{W}_Q^{(k)}, \mathbf{W}_K^{(k)}, \mathbf{W}_V^{(k)} \in \mathbb{R}^{d_h \times d}$ are learnable projections for head $k$.

The attention score (or embedding) for edge $(i,j) \in E_t$ is:

$$\mathbf{s}_{ij}^{(k)} = \alpha_{ij}^{(k)} \cdot \mathbf{v}_{ij}^{(k)} \tag{A5}$$

where the attention weight is:

$$\alpha_{ij}^{(k)} = \frac{\exp\left(\frac{\left(\mathbf{q}_i^{(k)}\right)^\top \mathbf{k}_j^{(k)}}{\sqrt{d_h}}\right)}{\sum\limits_{(i,l)\in E_t} \exp\left(\frac{\left(\mathbf{q}_i^{(k)}\right)^\top \mathbf{k}_l^{(k)}}{\sqrt{d_h}}\right)} \tag{A6}$$

The final multi-head edge representation is:

$$\hat{\mathbf{e}}_{ij} = \big\|_{k=1}^{K} \mathbf{s}_{ij}^{(k)} \tag{A7}$$

**Edge Score Refinement via Message Passing**  We refine edge scores using a message passing layer over the graph structure. For each edge $(i, j) \in E_t$, we define:

$$\hat{\mathbf{s}}_{ij} = \phi\left(\mathbf{s}_{ij},\ \square_{(u,v)\in N_{ij}}\psi\left(\mathbf{x}_u, \mathbf{x}_v, \mathbf{s}_{uv}\right)\right) \tag{A8}$$

- $\psi$: message function that combines sender/receiver node features $\mathbf{x}_u, \mathbf{x}_v$ and current edge score $\mathbf{s}_{uv}$, e.g.:

$$\psi(\mathbf{x}_u, \mathbf{x}_v, \mathbf{s}_{uv}) = \text{MLP}([\mathbf{x}_u \,\|\, \mathbf{x}_v \,\|\, \mathbf{s}_{uv}]) \tag{A9}$$

### A.2 *Disc* ARCHITECTURE

Our proposed architecture for $Disc$ shares the same structure as $f_\theta$. As shown in Fig. A1, to compute the score guidance, we first obtain the representations $\hat{\mathbf{X}}_t, \hat{\mathbf{F}}_t$, and $\hat{\mathbf{e}}_t$ based on the input node features $\mathbf{X}_t$, edge features $\mathbf{F}_t$, and normalized edge labels $\mathbf{e}$. Then, in $Disc$, we apply a softmax operation to each component as follows:

$$\mathbf{S_H} = \text{softmax}(\hat{\mathbf{X}}_t), \quad \mathbf{S_F} = \text{softmax}(\hat{\mathbf{F}}_t), \quad \mathbf{s_e} = \text{softmax}(\hat{\mathbf{e}}_t) \tag{A10}$$

### A.3 COMPUTATIONAL COMPLEXITY

Since we employ a GNN-based model for the denoising process, the computational complexity of the denoising core, $f_\theta$, is linear with respect to the number of edges, i.e., $O(|E|)$. In addition, the edge denoising step, described in equation 4, has a complexity of $O(|E_t|)$ at each time step $t$. The discriminator component, $\text{GNN}_\beta$, has the same complexity as $f_\theta$, namely $O(|E|)$. Both node and edge feature denoising—guided by the discriminator and defined in equations 9 and 10, together require $O(2 \times |E|)$ operations. Therefore, the overall computational complexity of our proposed model remains linear with respect to the input graph size.

### A.4 IMPLEMENTATION DETAILS

We set 256 time steps for the diffusion forward process, using a cosine noise schedule for $\alpha$. For $f_\theta = \text{GNN}_\theta$ and $Disc = \text{GNN}_\beta$, we use a 5-layer GIN Xu et al. (2018) and a 3-layer GCN (Kipf & Welling, 2017), respectively, both with a hidden dimension of 64. Models are optimized using the Adam optimizer, with learning rates of 0.001 for $\text{GNN}_\theta$ and 0.0001 for $\text{GNN}_\beta$, over 10,000 training epochs. For simplicity, we assume a constant $\rho_t^{\min}$, defined as the ratio of the number of nodes to the number of edges in the input graph.

## B  NOISE INJECTION VIA STOCHASTIC EDGE DELETION

At each diffusion step $t$, the forward process applies a stochastic deletion operator $\mathcal{D}_t$ that randomly removes edges from the previous graph, parametrized by the step-specific deletion probability $\epsilon_t$:

$$E_{t+1} = \mathcal{D}_t(E_t, \epsilon_t),$$

where each existing edge in $E_t$ is independently deleted with probability $\epsilon_t$:

$$q_e(e_{uv}^{(t+1)} = 0 \mid e_{uv}^{(t)} = 1) = \epsilon_t, \qquad q_e(e_{uv}^{(t+1)} = 1 \mid e_{uv}^{(t)} = 1) = 1 - \epsilon_t,$$

and non-edges remain unchanged, i.e., $q_e(e_{uv}^{(t+1)} = 0 \mid e_{uv}^{(t)} = 0) = 1$.

Since each deletion step is an independent Bernoulli trial applied to every edge, this iterative process admits a closed-form expression conditioned on the original graph $E_0$.

The probability that an edge from the original graph $E_0$ *survives* all $t$ deletion steps is:

$$\prod_{i=1}^{t}(1 - \epsilon_i).$$

This leads to the **total removal probability** $\eta_t$:

$$\eta_{\epsilon_t} = 1 - \prod_{i=1}^{t}(1 - \epsilon_i).$$

Thus, the noised graph $E_t$ can be viewed as the result of a single Bernoulli trial applied directly to $E_0$. The full forward transition probability $q_e(E_t \mid E_0)$ is a product of independent Bernoulli distributions over all possible edges, conditioned on $E_0$:

$$q_e(E_t \mid E_0) = \prod_{(u,v)} q_e(e_{uv}^{(t)} \mid e_{uv}^{(0)}).$$

For any edge $(u, v)$, the conditional probability $q_e(e_{uv}^{(t)} = 1 \mid e_{uv}^{(0)})$ is given by:

$$q_e(e_{uv}^{(t)} = 1 \mid e_{uv}^{(0)}) = e_{uv}^{(0)} \cdot (1 - \eta_{\epsilon_t}).$$

This formulation shows that edge deletion depends solely on the global noise schedule—not on local connectivity or graph structure.

## C    PROOF OF COROLLARY 3.1

**Neighborhood Size Requirement**    Let $\Omega$ denote the minimum required size of a node $v$'s $L$-hop neighborhood to enable accurate local structure reconstruction by the GNN:

$$|B_L^{(t)}(v)| \geq \Omega. \tag{A11}$$

**Expected Neighborhood Size After Edge Deletion**    Let $d_{\text{org}}$ denote the average degree of the original graph and let $\rho_t = 1 - \epsilon_t$ be the probability of edge preservation. The expected degree after deletion is:

$$d_t = \rho_t \cdot d_{\text{org}}. \tag{A12}$$

Assuming tree-like expansion in the $L$-hop neighborhood, we approximate:

$$\mathbb{E}[|B_L^{(t)}(v)|] \approx \sum_{\ell=0}^{L} (\rho_t d_{\text{org}})^{\ell} = \frac{(\rho_t d_{\text{org}})^{L+1} - 1}{\rho_t d_{\text{org}} - 1}. \tag{A13}$$

We require this to be at least $\Omega$:

$$\frac{(\rho_t d_{\text{org}})^{L+1} - 1}{\rho_t d_{\text{org}} - 1} \geq \Omega. \tag{A14}$$

**Lower Bound Approximation for Edge Density** To simplify, for sufficiently large $L$ and $\rho_t d_{\text{org}} > 1$, we approximate the constraint by the dominant term:

$$(\rho_t d_{\text{org}})^L \geq \Omega \quad \Rightarrow \quad \rho_t \geq \left(\frac{\Omega}{d_{\text{org}}^L}\right)^{1/L}. \tag{A15}$$

This yields a lower bound on edge density required to ensure sufficient neighborhood size:

$$\rho_t^{\min} \geq \left(\frac{\Omega}{d_{\text{org}}^L}\right)^{1/L} \tag{A16}$$

# D    PROOF OF PROPOSITION 3.1

Let $\mathbf{h}_v^{(L)}$ be the embedding of node $v$ at layer $L$, and consider the Jacobian:

$$J_{vu}^{(L)} := \frac{\partial \mathbf{h}_v^{(L)}}{\partial \mathbf{h}_u^{(0)}}, \tag{A17}$$

which quantifies the influence of node $u$'s initial features on node $v$'s final representation.

Then:

1. The Jacobian entry $J_{vu}^{(L)}$ can be expressed as a sum over all effective computation paths of length $L$ from $u$ to $v$:

$$J_{vu}^{(L)} = \sum_{p \in \mathcal{P}_{u \to v}^{(L)}} \prod_{\ell=1}^{L} \frac{\partial \mathbf{h}_{i_\ell}^{(\ell)}}{\partial \mathbf{h}_{i_{\ell-1}}^{(\ell-1)}}, \tag{A18}$$

   where $Path_{u \to v}^{(L)}$ is the set of active paths from $u$ to $v$ in $L$ steps, and each path $p = (i_0 = u, i_1, \ldots, i_L = v)$ must follow preserved edges in the graph.

2. The expected number of active paths is upper bounded by:

$$\mathbb{E}[|Path_{u \to v}^{(L)}|] \leq (\rho_t d_{\text{org}})^L. \tag{A19}$$

3. Therefore, when $\rho_t < \left(\frac{\Omega}{d_{\text{org}}^L}\right)^{1/L}$, the expected $L$-hop neighborhood size of a node $v$ is smaller than the minimum required size $\Omega$, and the number of active paths decays exponentially with $L$.

4. Consequently, $J_{vu}^{(L)} \to 0$ for many node pairs $(u, v)$, meaning the GNN cannot propagate or differentiate structural information beyond $L$-hops effectively.

5. This phenomenon corresponds to *over-squashing*, where too many distant messages are compressed into limited embeddings, resulting in poor representation quality and inaccurate graph recovery.

# E    THEORETICAL VIEW OF PROPOSITION 3.2

We present the proof with respect to $\mathbf{X}$, representing node features. Similar reasoning extends to edge features and structure under appropriate continuity assumptions.

Let $p_\theta(\mathbf{X}_t) \in \mathcal{P}_2(\mathbb{R}^{d_v})$ be the distribution of $\mathbf{X}$ at time step $t$, where

$$\mathcal{P}_2(\mathbb{R}^{d_v}) = \left\{\mu \in \mathcal{P}(\mathbb{R}^{d_v}) \mid \int_{\mathbb{R}^{d_v}} \|x\|^2 \, d\mu(x) < \infty\right\}$$

is the space of probability measures with finite second moment.

Consider the KL divergence as a functional relative to a target distribution $p_{\mathbf{x}_0}$:

$$\mathcal{F}(p_\theta(\mathbf{X}_t)) = \mathrm{KL}(p_\theta(\mathbf{X}_t) \,\|\, p_{\mathbf{x}_0}) = \int p_\theta(\mathbf{X}_t) \log \frac{p_\theta(\mathbf{X}_t)}{p_{\mathbf{x}_0}(\mathbf{X}_t)} \, d\mathbf{X}_t.$$

The Wasserstein gradient flow of $\mathcal{F}$ follows the PDE:

$$\frac{\partial p_\theta(\mathbf{X}_t)}{\partial t} = \nabla \cdot \left( p_\theta(\mathbf{X}_t) \nabla \log \frac{p_\theta(\mathbf{X}_t)}{p_{\mathbf{x}_0}(\mathbf{X}_t)} \right),$$

which corresponds to a sample-wise update rule (interpreted in a Lagrangian particle framework) as:

$$\frac{d\mathbf{X}_t}{dt} = -\nabla_{\mathbf{X}_t} \log p_\theta(\mathbf{X}_t) + \nabla_{\mathbf{X}_t} \log p_{\mathbf{x}_0}(\mathbf{X}_t).$$

In diffusion models that follow the score matching approach—and more broadly, in other diffusion-based models such as denoising diffusion probabilistic models and denoising diffusion implicit models, when no additional assumptions are made—the reverse process is defined using the score function:

$$\frac{d\mathbf{X}_t}{dt} = -\nabla_{\mathbf{X}_t} \log p_\theta(\mathbf{X}_t),$$

which approximates the drift induced by the learned marginal at time $t$.

Now, let the discriminator $D_x(\mathbf{X}_t)$ be trained to distinguish between real samples $\mathbf{X}_t \sim p_{\mathbf{x}_0}$ and generated samples $\mathbf{X}_t \sim p_\theta$. Assuming it is trained to optimality with a binary cross-entropy loss and equal class priors, the optimal discriminator satisfies:

$$D_x^*(\mathbf{X}_t) = \frac{p_\theta(\mathbf{X}_t)}{p_{\mathbf{x}_0}(\mathbf{X}_t) + p_\theta(\mathbf{X}_t)}.$$

Taking the logarithm and gradient of the optimal discriminator gives an estimate of the log-density ratio:

$$\nabla_{\mathbf{X}_t} \log D_x^*(\mathbf{X}_t) \approx \nabla_{\mathbf{X}_t} \log \left( \frac{p_\theta(\mathbf{X}_t)}{p_{\mathbf{x}_0}(\mathbf{X}_t)} \right) = \nabla_{\mathbf{X}_t} \log p_\theta(\mathbf{X}_t) - \nabla_{\mathbf{X}_t} \log p_{\mathbf{x}_0}(\mathbf{X}_t).$$

Consider the discriminator-guided update:

$$\frac{d\mathbf{X}_t}{dt} = -\nabla_{\mathbf{X}_t} \log p_\theta(\mathbf{X}_t) + \lambda \cdot \left( \nabla_{\mathbf{X}_t} \log p_\theta(\mathbf{X}_t) - \nabla_{\mathbf{X}_t} \log p_{\mathbf{x}_0}(\mathbf{X}_t) \right),$$

which simplifies to:

$$\frac{d\mathbf{X}_t}{dt} = -(1 - \lambda)\nabla_{\mathbf{X}_t} \log p_\theta(\mathbf{X}_t) - \lambda \nabla_{\mathbf{X}_t} \log p_{\mathbf{x}_0}(\mathbf{X}_t).$$

As $\lambda \to 1$, the update becomes:

$$\frac{d\mathbf{X}_t}{dt} = -\nabla_{\mathbf{X}_t} \log p_{\mathbf{x}_0}(\mathbf{X}_t),$$

which is the gradient flow that transports samples toward the data distribution $p_{\mathbf{x}_0}$.

## F  DISCUSSION OF COROLLARY 3.2

The Weisfeiler-Lehman (WL) test iteratively refines node labels based on their neighborhood. After $L_L$ iterations, two initially indistinguishable graphs may become distinguishable if their refined node labels differ.

A GNN with $L$ layers follows a similar neighborhood aggregation scheme, where each node aggregates features from its $L_L$-hop neighbors and available information for such $v$ is $B_{L_L}(v) = \{u \in$

$V \mid \text{dist}(u, v) \leq L_L\}$. If a GNN has fewer than $L_L$ layers, it cannot fully capture the $L$-hop structure necessary to separate graphs requiring $L$ iterations of WL.

Thus, for a GNN to match the expressiveness of the WL test, it must have at least:

$$L \geq L_L. \tag{A20}$$

This establishes the lower bound on the number of GNN layers necessary to achieve the expressiveness of the WL test.

## G  DISCUSSION OF COROLLARY 3.3

In an undirected graph, information propagates bidirectionally. This means that in $L$ layers, a node aggregates information from its $2L$-hop neighborhood.

For a node to access information from all other nodes, it must be able to reach any node within at most $D$ hops. Since information spreads both ways, the minimum required layers are:

$$L \geq \frac{D}{2}. \tag{A21}$$

If $L < \frac{D}{2}$, then at least one node cannot access the entire graph, making discrimination between real and unrealistic graphs under incomplete node information challenging.

In a directed graph, information propagates only in the direction of edges. This means that in $L$ layers, a node aggregates information only from nodes within its $L$-hop directed neighborhood.

In the worst-case scenario, there exists a node $u$ that requires $D_{\text{dir}}$ directed steps to reach node $v$. To ensure complete propagation of information, each node must receive information from every other node from which it is reachable.

Since message passing follows directed edges, we require:

$$L \geq D_{\text{dir}}. \tag{A22}$$

If $L < D_{\text{dir}}$, then at least one node cannot access the entire graph, preventing full discrimination of real and unrealistic graphs.

## H  EXTENDED RESULTS

As mentioned earlier, we evaluate the models after generating 32 samples per graph. Table A2 reports the average MMD results in the models, quantifying the distributional distance between the generated graphs and their input graphs in terms of degree, clustering coefficient and node or edge features. As shown, models like GraphMaker, which rely heavily on node features, perform better in feature generation for datasets such as Citeseer with rich node attributes, but perform less effectively in structure generation.

Table A1: Small dataset statistics.

| Dataset | nodes | edges | graphs | feature |
|---------|-------|-------|--------|---------|
| Community | [60, 160] | [231, 1,965] | 510 | – |
| Ego | [50, 399] | [57, 1,071] | 757 | – |

Beyond distribution-level evaluation, Table A3 presents statistics comparing a randomly selected example from the 32 generated graphs with its original input counterpart. These statistics include the average ($AVG_d$), maximum ($MAX_d$), and variance($VAR_d$) of node degrees, as well as PLE and CPL.

The results in Table A3 should be interpreted carefully. A higher maximum degree may be desirable, but only in the context of the corresponding average degree. For instance, when the average degree is relatively low and the maximum degree is high, the elevated maximum degree may suggest greater

Table A2: MMD scores ($\times 10^{-2}$) for degree distribution, clustering coefficient, node features, and edge features across datasets. A '-' indicates the model is either incapable of generating that feature or the dataset does not contain it. Lower scores are better; best and second-best results are highlighted in **bold** and underlined, respectively.

| Method | Cora | | | | Citeseer | | | | Pubmed | | | |
| --- | --- | --- | --- | --- | --- | --- | --- | --- | --- | --- | --- | --- |
| | Deg. | Clus. | NodeF | EdgeF | Deg. | Clus. | NodeF | EdgeF | Deg. | Clus. | NodeF | EdgeF |
| GraphRNN | 21.83±0.80 | 4.79±0.25 | - | - | **2.01**±0.14 | 1.50±0.10 | - | - | 15.00±0.62 | 24.00±1.10 | - | - |
| GraphMaker | 13.21±0.55 | 9.18±0.41 | **0.63**±0.05 | - | 17.50±0.70 | 23.50±1.00 | **0.17**±0.01 | - | 16.20±0.80 | 34.60±1.30 | 8.93±0.50 | - |
| EDGE | 5.75±0.20 | **4.47**±0.18 | - | - | 4.78±0.19 | 1.33±0.09 | - | - | 9.91±0.32 | 1.27±0.07 | - | - |
| ARROW-Diff | 5.65±0.22 | 4.73±0.21 | - | - | 4.13±0.16 | **1.12**±0.07 | - | - | **9.88**±0.30 | 3.30±0.18 | - | - |
| DGDGL (ours) | **5.45**±0.19 | 4.53±0.17 | 0.75±0.06 | - | 3.97±0.15 | 1.32±0.09 | 0.21±0.02 | - | 9.89±0.31 | **0.29**±0.02 | **8.05**±0.42 | - |

| Method | Alpha | | | | OTC | | | | Elliptic | | | |
| --- | --- | --- | --- | --- | --- | --- | --- | --- | --- | --- | --- | --- |
| | Deg. | Clus. | NodeF | EdgeF | Deg. | Clus. | NodeF | EdgeF | Deg. | Clus. | NodeF | EdgeF |
| GraphRNN | 9.87±0.40 | 69.80±2.50 | - | - | 9.95±0.35 | 7.81±0.28 | - | - | **1.19**±0.08 | 1.73±0.09 | - | - |
| GraphMaker | 25.70±0.90 | 34.20±1.20 | - | - | 53.10±1.50 | 34.70±1.00 | - | - | 10.10±0.62 | 10.80±0.48 | 1.18±0.09 | - |
| EDGE | 5.01±0.19 | 2.73±0.11 | - | - | 5.31±0.21 | 3.11±0.12 | - | - | 6.19±0.23 | 2.07±0.09 | - | - |
| ARROW-Diff | 5.03±0.20 | 2.75±0.12 | - | - | 5.23±0.19 | 2.77±0.11 | - | - | 6.17±0.22 | 1.01±0.06 | - | - |
| DGDGL (ours) | **4.67**±0.17 | **2.47**±0.10 | - | **4.30**±0.15 | **4.67**±0.18 | **2.01**±0.09 | - | **5.50**±0.17 | 6.28±0.25 | **0.97**±0.05 | **0.14**±0.02 | **5.78**±0.19 |

structural diversity. This typically leads to higher degree variance, reflecting increased diversity in node degrees and, ultimately, in the overall graph structure.

Additionally, EO provides insight into whether a model is learning generalizable patterns or merely memorizing the input distribution. While models such as ARROW-Diff show acceptable performance, their high edge overlap with the input graph raises concerns about memorization. This tendency often results in lower degree variance and reduced generalization capability.

Furthermore, a higher average degree often corresponds to a shorter characteristic path length (CPL), as nodes are more densely connected. When we consider the average, maximum, and variance of node degrees jointly, our model demonstrates greater flexibility, allowing it to generate more diverse graph structures while generalizing from the input graph. Additionally, PLE reflects how well the model preserves the scale-free nature of the input graph. Fig. A2 illustrates one of the 32 generated examples compared to its corresponding input graph for each dataset, showing that our model closely follows the input distribution. Our model achieves the best performance across all datasets in terms of PLE, indicating strong alignment with power-law degree distributions. For directed datasets like Elliptic and OTC, distributional and statistical evaluations are performed based on node in-degree. Figure A3 shows the generated edge attributes, where timestamps are decomposed into month, day, and hour. As illustrated, our model generates edge attributes that closely match the original features.

Table A3: Statistics of the generated graphs compared to their input counterparts. For PLE and CPL, values closer to the ground truth are better. EO for the ground truth refers to its overlap with itself. Best and second-best results are highlighted in **bold** and underlined, respectively.

| | Cora | | | | | | Citeseer | | | | | | Pubmed | | | | | |
| --- | --- | --- | --- | --- | --- | --- | --- | --- | --- | --- | --- | --- | --- | --- | --- | --- | --- | --- |
| | EO↓ | PLE | CPL | $AVG_d$↓ | $MAX_d$↑ | $VAR_d$↑ | EO↓ | PLE | CPL | $AVG_d$↓ | $MAX_d$↑ | $VAR_d$↑ | EO↓ | PLE | CPL | $AVG_d$↓ | $MAX_d$↑ | $VAR_d$↑ |
| ground truth | 100 | 1.83 | 6.31 | 3.89 | 168 | 27.3 | 100 | 1.88 | 9.32 | 2.73 | 99 | 11.4 | 100 | 2.21 | 6.33 | 4.49 | 171 | 55.2 |
| GraphRNN | 0.01 | 1.9 | **5.39** | **1.79** | 20 | 2.31 | 0.1 | 2.3 | **10.1** | **2.08** | 13 | 2.19 | 0.1 | 1.78 | 4.3 | **5.1** | 15 | 8.6 |
| GraphMaker | 3.7 | 1.3 | 2.64 | 8.8 | 25 | 17.8 | 5.3 | 0.97 | 1.23 | 8.78 | 30 | 8.65 | 10.7 | 1.3 | 2.74 | 12.4 | 30 | 17.6 |
| EDGE | 2.1 | 1.76 | 4.98 | 7.63 | 87 | 21.8 | 6.1 | 1.72 | 6.41 | | **84** | 10.3 | 8.2 | 2.15 | **6.23** | 8.3 | 91 | 36.8 |
| ARROW-Diff | 41.2 | 1.66 | 4.82 | 5.73 | 84 | 13.7 | 64.5 | 1.77 | 5.43 | 4.27 | 68 | 13.2 | 51.3 | 1.89 | 5.27 | 6.01 | **147** | 39.3 |
| DGDGL (ours) | 0.0 | **1.82** | 4.84 | 6.38 | **93** | **35.6** | 0.0 | **1.9** | 5.52 | 4.03 | 71 | **51.4** | 0.0 | **2.19** | 5.58 | 5.7 | 145 | 47.4 |

| | Alpha | | | | | | OTC | | | | | | Elliptic | | | | | |
| --- | --- | --- | --- | --- | --- | --- | --- | --- | --- | --- | --- | --- | --- | --- | --- | --- | --- | --- |
| | EO↓ | PLE | CPL | $AVG_d$↓ | $MAX_d$↑ | $VAR_d$↑ | EO↓ | PLE | CPL | $AVG_d$↓ | $MAX_d$↑ | $VAR_d$↑ | EO↓ | PLE | CPL | $AVG_d$↓ | $MAX_d$↑ | $VAR_d$↑ |
| ground truth | 100 | 1.24 | 3.57 | 7.46 | 510 | 401.95 | 100 | 1.24 | 3.57 | 7.3 | 795 | 530.7 | 100 | 2.08 | 242 | 2.28 | 163 | 7.19 |
| GraphRNN | 0.1 | 0.88 | 1.71 | **3.7** | 7.1 | 8.1 | 0.0 | 0.68 | 1.73 | **3.5** | 11 | 7.08 | 1.6 | 1.26 | 7.8 | 2.22 | 27 | 5.8 |
| GraphMaker | 9.2 | 0.35 | 1.37 | 15.6 | 70 | 23.4 | 11.8 | 0.43 | 1.37 | 15.1 | 67 | 40.7 | 9.3 | 1.1 | 3.1 | 7.4 | 93 | 6.5 |
| EDGE | 3.7 | 0.91 | 2.1 | 11.3 | 283 | 57.8 | 5.6 | 0.87 | 1.9 | 11.5 | 342 | 83.5 | 3.1 | 1.27 | 5.83 | 5.4 | 107 | 11.7 |
| ARROW-Diff | 67.8 | 0.99 | 2.05 | 8.1 | 113 | 23.6 | 53.7 | 0.82 | 2.02 | 9.7 | 115 | 21.3 | 61.3 | 1.32 | 6.1 | 4.5 | 103 | 10.2 |
| DGDGL (ours) | 0.0 | **1.29** | 3.65 | 7.37 | **300** | **123.4** | 0.0 | **1.39** | 3.62 | 9.1 | 377 | **131.5** | 0.0 | **1.71** | 8.4 | **2.18** | **219** | **17.64** |

**Results for smaller graphs** As an additional evaluation track to assess our proposed method, we consider two small datasets—Community and Ego (You et al., 2018)—with statistics provided in Table A1. In addition to the MMD of clustering coefficient and node degree distributions, we also consider Orbit MMD, which is designed to evaluate graph motifs (Hocevar & Demsar, 2014). We report the FID (Thompson et al., 2022) to assess the overall quality of the generated graphs.

Furthermore, we assess uniqueness using the Weisfeiler–Lehman (WL) test on the generated graphs and evaluate novelty by checking isomorphism against the 30% of input graphs held out as the test set. The results in Table A4 are averaged over 32 generated samples per dataset. As shown in the table, while models such as GraphMaker exhibit a substantial performance drop on smaller graphs,

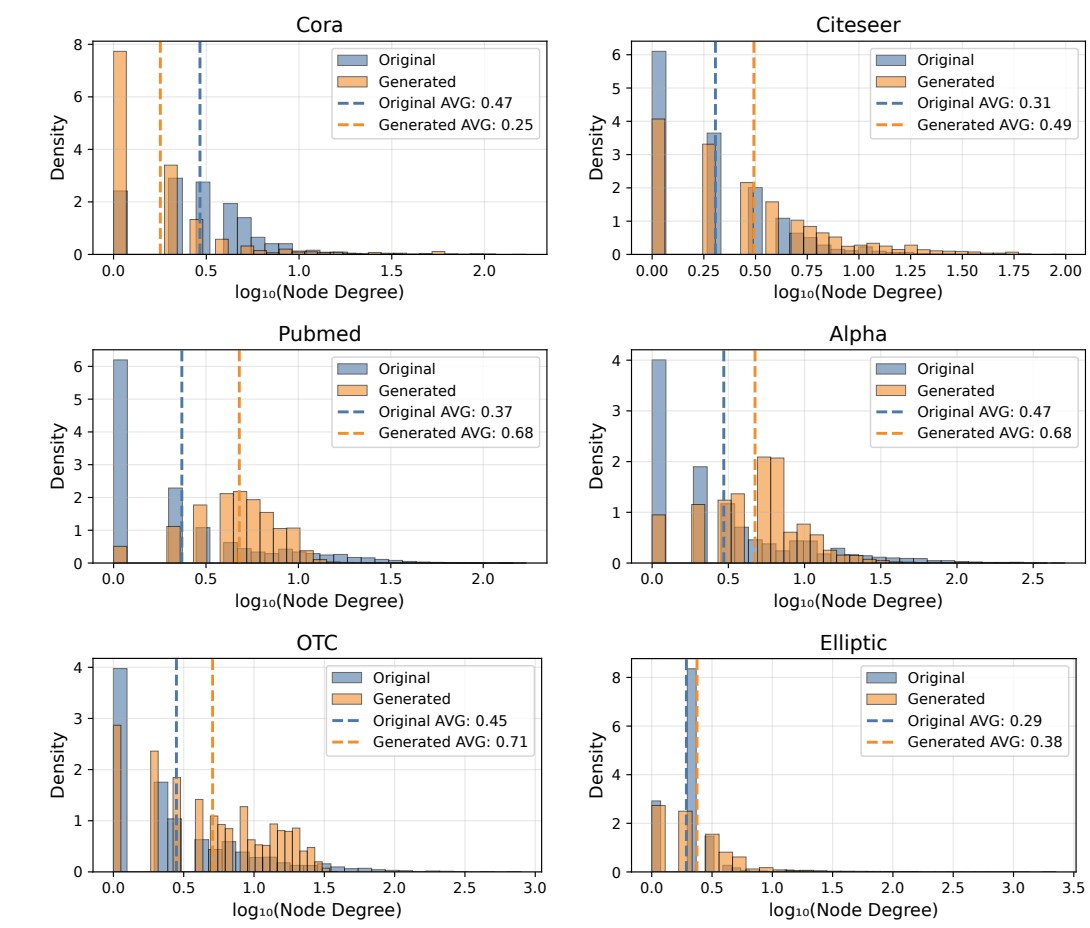

Figure A2: Node degree comparison between generated graph and input graphs across six datasets.

Table A4: Comparison of metrics for small graphs. MMD values are reported as $\times 10^{-2}$ (lower is better); FID↓ is based on a neural feature space; *Uniq*↑ indicates the percentage of non-isomorphic generated graphs; and *Nov*↑ measures the percentage of generated graphs not present in the training set (both higher are better).

| | Community | | | | | | Ego | | | | | |
|---|---|---|---|---|---|---|---|---|---|---|---|---|
| | **MMD** ($\times 10^{-2}$)↓ | | | FID↓ | Uniq↑ | Nov↑ | **MMD** ($\times 10^{-2}$)↓ | | | FID↓ | Uniq↑ | Nov↑ |
| Method | Deg. | Clus. | Orb. | | (%) | (%) | Deg. | Clus. | Orb. | | (%) | (%) |
| GraphRNN | 15.23 ±0.41 | 8.53 ±0.29 | 2.97 ±0.18 | 8.38 ±0.22 | 72.4 ±1.8 | 68.9 ±2.1 | 7.88 ±0.37 | 101.45 ±2.4 | 14.87 ±0.53 | 87.76 ±0.74 | 65.3 ±2.1 | 62.4 ±2.3 |
| GraphMaker | 27.88 ±0.95 | 32.33 ±1.07 | 19.78 ±0.66 | 56.11 ±1.42 | 45.8 ±1.3 | 43.1 ±1.6 | 35.78 ±1.22 | 33.60 ±0.91 | 67.54 ±1.84 | 189.95 ±2.3 | 38.4 ±0.8 | 35.9 ±1.1 |
| DiGress | 3.07 ±0.11 | **1.69** ±0.06 | 2.89 ±0.08 | 4.07 ±0.09 | 98.6 ±0.3 | 98.1 ±0.5 | 6.09 ±0.15 | **0.97** ±0.05 | 11.09 ±0.22 | 16.76 ±0.31 | 99.1 ±0.1 | 97.7 ±0.2 |
| EDGE | 1.63 ±0.05 | 7.01 ±0.27 | 2.10 ±0.04 | 2.24 ±0.06 | **99.9** ±0.0 | 98.8 ±0.1 | 5.59 ±0.09 | 16.61 ±0.48 | 6.21 ±0.05 | 14.59 ±0.12 | 99.5 ±0.1 | 98.6 ±0.1 |
| ARROW-Diff | 12.38 ±0.46 | 86.49 ±2.5 | 16.46 ±0.67 | 13.43 ±0.41 | 41.25 ±1.9 | 39.8 ±1.7 | 16.13 ±0.38 | 12.88 ±0.34 | 15.23 ±0.72 | 22.45 ±0.59 | 48.7 ±1.2 | 45.5 ±1.4 |
| DGDGL (ours) | **1.09** ±0.03 | 7.84 ±0.18 | 21.31 ±0.74 | **1.85** ±0.02 | **99.9** ±0.0 | 98.3 ±0.1 | 35.55 ±0.91 | 4.29 ±0.09 | 11.05 ±0.31 | **4.34** ±0.05 | **100.0** ±0.0 | **99.0** ±0.0 |

our model maintains performance comparable to small-graph–oriented methods like DiGress across several key metrics.

Moreover, due to the incorporation of noise injection into the structure—via edge deletion and the enforcement of a minimum edge density—our model consistently generates new graph samples rather than memorizing the training data, unlike models such as ARROW-Diff.

**Empirical sampling time** As we illustrated earlier (Appendix A), our proposed model has linear complexity with respect to the number of edges. Here, we examine this behavior in practice.

We measure the sampling (generation) time for graphs of different sizes and compare it across models. As shown in Figure A6, our proposed model achieves reasonable sampling times even for large graphs—requiring less than 110 seconds for graphs with more than 10k nodes—whereas

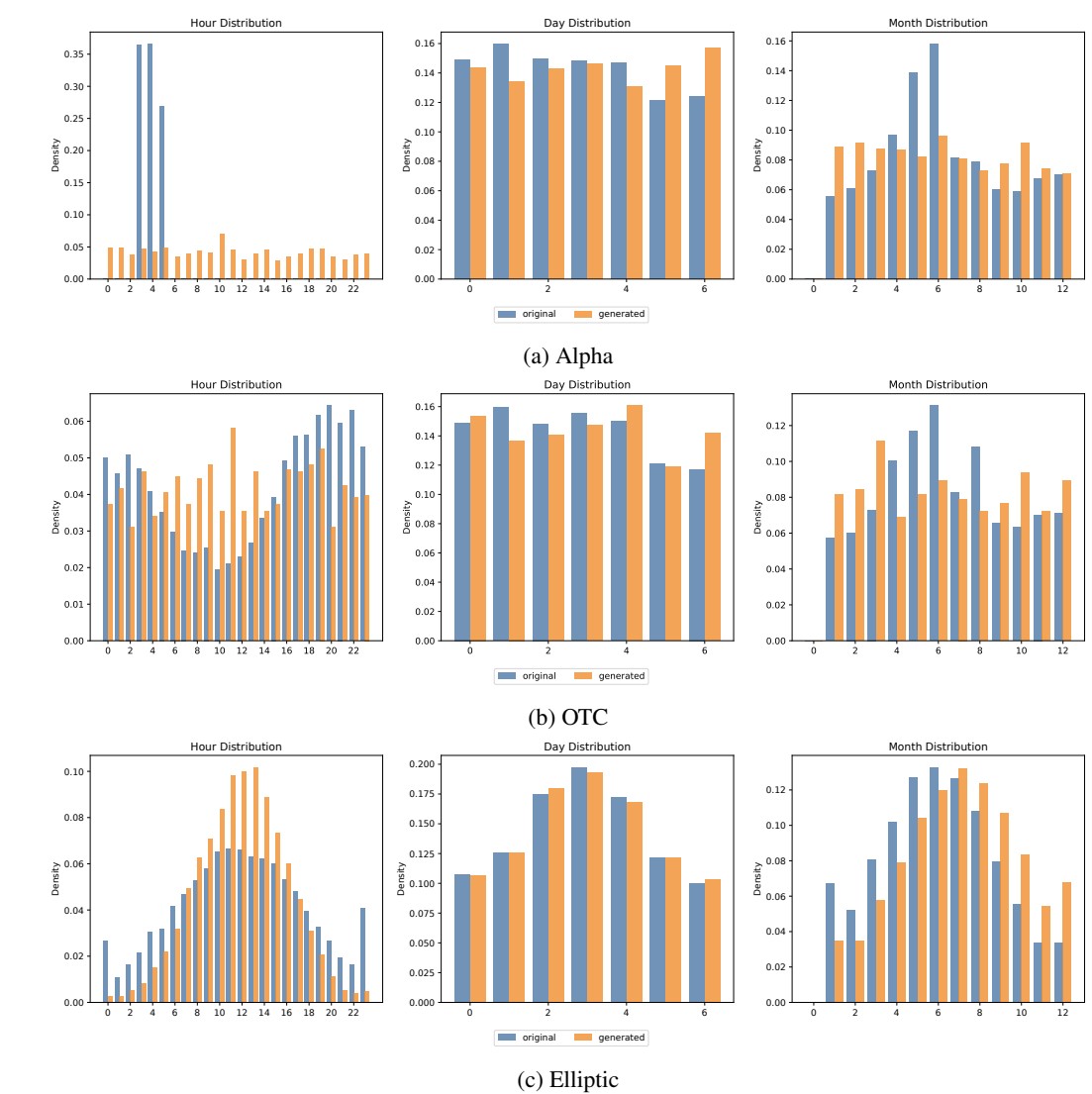

Figure A3: Comparison of generated and original edge features for directed datasets.

models such as EDGE and GraphMaker require around 200 seconds on average. ARROW-Diff achieves the best sampling time overall for larger graphs.

The results in Figure A6 were obtained by averaging over 16 sampling runs for each model under identical conditions, including the same hidden dimensions, random seeds, and other settings, using an NVIDIA GeForce RTX 3090.

**Anomaly detection**  To evaluate the discriminator's performance in anomaly detection, we focus on the Alpha and OTC datasets (Fig. A4a), where edge labels range from –10 to +10. As demonstrated in Fig. A4b, ($Disc$) assigns higher anomaly scores to edges with negative labels compared to the average edge anomaly score. This indicates that classifying edges based on their anomaly scores enables the discriminator to accurately detect the majority of negatively labeled (i.e., anomalous) edges. Additionally, we compare the output of $Disc$ with a baseline GNN model, denoted as $GNN_{base}$, which shares the same architecture but is trained from scratch for an edge regression task. Formally, $GNN_{base}$ is defined as:

$$\text{GNN}_{\text{base}} \leftarrow \text{Disc},$$
$$\mathcal{L}_{D_{base}} = \left\| \hat{\mathbf{s}}_e^{base} - \mathbf{e}_0 \right\|^2 \tag{A23}$$

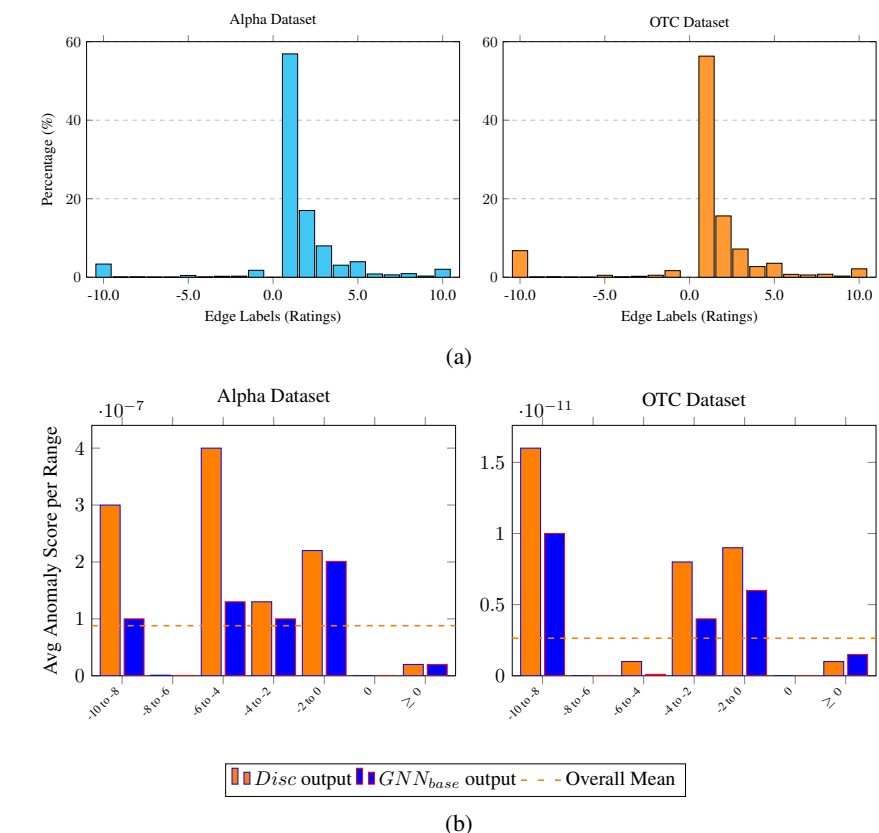

(a)

(b)

Figure A4: (a) Edge label distributions for the Alpha and OTC datasets. (b) Average anomaly scores across edge-label ranges, with dashed lines indicating the mean scores produced by $Disc$.

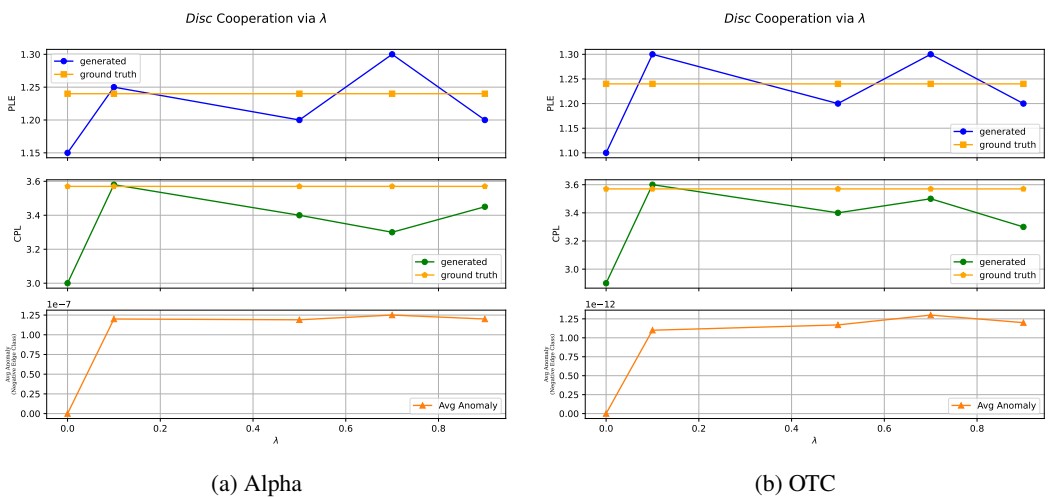

(a) Alpha                    (b) OTC

Figure A5: Disc cooperation via $\lambda$. As shown, increasing $\lambda$ beyond 0.5 does not necessarily improve generation performance, but it leads to higher anomaly scores for negative edge classes, which are treated as anomalies. For both PLE and CPL metrics, values closer to those of the original input graph indicate better performance.

Fig. A4b shows both $\hat{\mathbf{s}}_e^{\text{base}}$ and $\mathbf{s}_e$, where $\mathbf{s}_e$ (produced by $Disc$) exhibits a stronger correlation with negatively labeled edges, indicating superior anomaly detection performance compared to the supervised baseline.

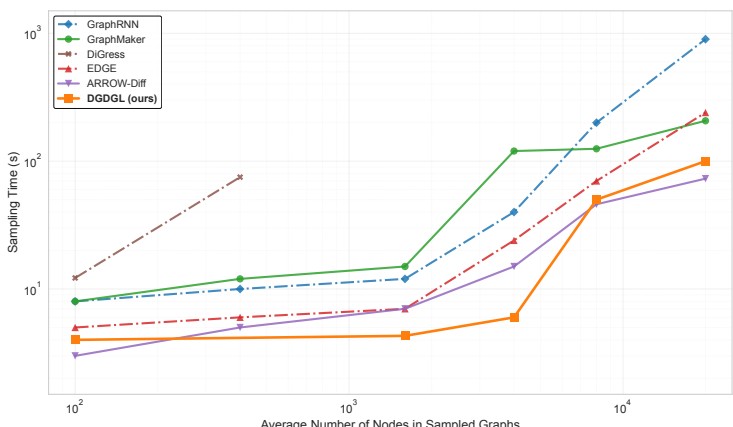

Figure A6: Sampling time across various graph sizes, averaged over 16 runs per model under identical conditions. The dash–dot line and solid line represent non-linear and linear models, respectively.

## H.1    ABLATION STUDY: **Disc** COOPERATION

According to our proposed method, an important question is how the discriminator ($Disc$) contributes to high-quality graph generation. To explore this, and with reference to equations 9 and 10, where the parameter $\lambda$ controls the strength of $Disc$ guidance, we test different values of $\lambda$ and evaluate performance based on three metrics: PLE, CPL, and the average anomaly score for negative edge classes.

From the results in Table A3, we observe that better PLE and CPL values generally correlate with improved overall generation performance. We focus on the Alpha and OTC datasets, as they allow us to analyze anomaly scores.

As shown in Fig. A5, incorporating $Disc$ guidance (i.e., using $\lambda > 0$) improves generation quality compared to no guidance ($\lambda = 0$). However, excessively strong guidance (e.g., $\lambda = 0.9$) does not further enhance generation performance. Interestingly, as $\lambda$ increases, the average anomaly score for negative edge classes in the original dataset also increases. This suggests that stronger discriminator participation enhances anomaly detection performance but may come at the cost of generation quality.

Therefore, we conclude that if the primary objective is anomaly detection rather than high-fidelity generation, stronger $Disc$ guidance may be beneficial. In contrast, balancing $\lambda$ is crucial when generation quality is the main goal, and anomaly detection is a secondary application.

## H.2    ABLATION STUDY: SATISFYING $\rho^{\min}$

As discussed, we apply structural noise via edge deletion to reduce computational overhead on large graphs. To maintain reconstructability, we enforce a minimum level of connectivity by randomly adding edges when deletion leads to sparse structures. Since estimating the optimal $\Omega$ is challenging (see Corollary 3.1), we approximate the minimum edge density at each step as $\rho_t^{\min} = \rho^{\text{org}}$, where $\rho^{\text{org}}$ is the original graph's edge density.

Fig. A7 presents the reconstruction loss for the Cora and OTC datasets. As depicted, the loss is lower when the edge addition mechanism is used to compensate for cases where edge deletion alone does not meet the minimum edge density requirement.

## I    DATASETS

Here we explain more details a bout datasets that have been used for evaluation.

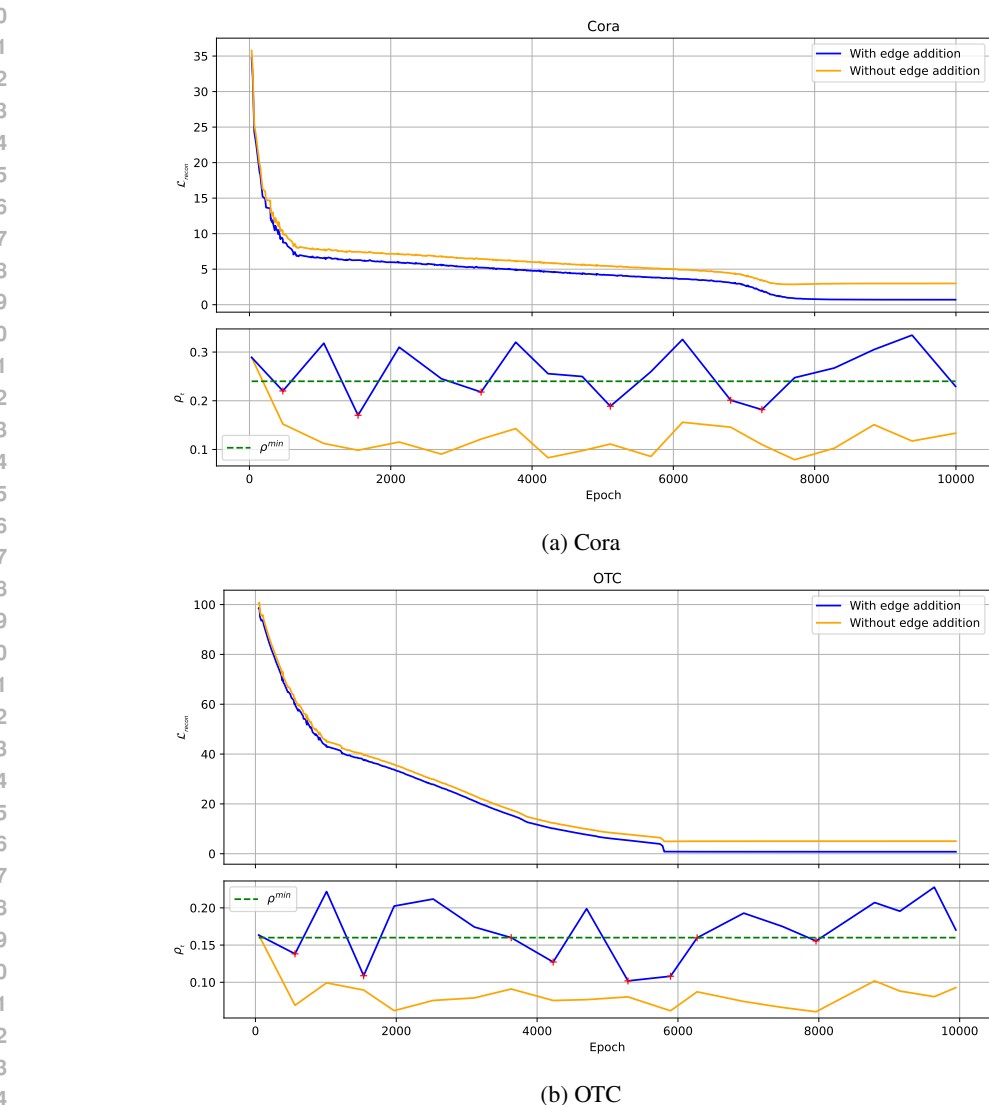

(a) Cora

(b) OTC

Figure A7: While edge addition generally introduces significant computational overhead, it can reduce reconstruction loss when used to meet the minimum edge density. Here, $\rho_t$ represents the edge density of the graph at each epoch during training.

**Citation networks** Citation networks, Cora, Citeseer, and Pubmed, comprise sparse bag-of-words feature vectors for each document, along with citation links representing the connections between documents (Sen et al., 2008).

**Signed Bitcoin Datasets: Alpha and OTC** The Alpha and OTC datasets are signed, directed networks representing who-trusts-whom relationships among users trading Bitcoin on two different platforms. Each edge is labeled with an integer from –10 (total distrust) to +10 (total trust), reflecting the level of trust between users. These networks are the first explicit weighted signed directed graphs made publicly available for research. Both datasets contain sparse graphs with similar structures; the primary differences lie in their sizes and the proportion of positive edges: Alpha has 89% positive edges, while OTC has 93% (Kumar et al., 2016; 2018).

**Elliptic** The Elliptic++ dataset contains approximately 203K Bitcoin transactions and 822K wallet addresses, enabling the detection of both fraudulent transactions and illicit actors through graph-based analysis (Elmougy & Liu, 2023b). Due to the large size of the dataset, we construct a subgraph

that spans all time steps of the original temporal graph. To ensure comprehensive temporal coverage while keeping the graph manageable, we apply overlapping community detection to select a representative subset of nodes across time. The resulting subgraph includes 18,945 nodes and 21,660 edges, with an average degree close to that of the original graph at each distinct time step. Furthermore, we convert each time step into multiple temporal resolutions—monthly, daily, and hourly—to explore the effects of time granularity on generation performance and graph dynamics.

**Community** This dataset consists of 500 two-community graphs with $60 \leq |N| \leq 160$. Each community is generated independently using the Erdős–Rényi (E-R) model (Erdos et al., 1960) with $n = |N|/2$ nodes and connection probability $p = 0.3$. Additionally, $0.05|N|$ inter-community edges are added between the two communities with uniform probability(You et al., 2018).

**Ego** This dataset contains 757 3-hop ego networks extracted from the Citeseer citation network (Sen et al., 2008), with $50 \leq |N| \leq 399$. Nodes represent documents, and edges represent citation relationships (You et al., 2018).

