# OpenReview forum: "Discriminator-Guided Diffusion for Generating Large Directed and Undirected Graphs"
_ICLR.cc/2026/Conference — Submitted to ICLR 2026_

### Official Review · Reviewer_QS4y · 2025-10-26

**Soundness:** 1
**Presentation:** 1
**Contribution:** 2
**Rating:** 2
**Confidence:** 4

**Summary:**

The paper proposes a scalable MPNN-based diffusion model for graph generation, combined with a discriminator for guidance, and extended to directed graphs.

**Strengths:**

1. The proposed discriminator design is an interesting idea in graph generative modeling.
2. The edge deletion strategy and the use of MPNNs with minimal edge number guarantee are reasonable and practically useful.

**Weaknesses:**

First, the paper motivation is not very clear. There are mainly 2 (actually 3) lines: one about scalability, one about guidance, and one about directed graph generation. However, they are not logically dependent on each other.
- Specifically, scalability is achieved by using GINs and claimed to be comparable to previous quadratic methods, but there is no comparison with important quadratic models such as DiGress, CatFlow, or DeFoG.
- The directed graph generation is a simple extension with no further justification, and current diffusion models can also support directed graph generation in a straightforward manner, which weakens the necessity and contribution of this part.

Second, from a presentation perspective, there are notations like $L_L$(Corollary 3.2), which is not very readable. Beyond that, in Table 3, there seem to be some mistakes. For instance, GraphRNN is better on Deg (Citeseer), Def (Elliptic), but the results of DGDGL (ours) are bolded. The columns including node features distance and edge feature distance that have no values. The 3 histograms in the paper are presented in 3 different formats (A2 A3 A4). The paper presentation is not complete.

Third, experimentally, as mentioned in the first point as well, the paper claims: “The results show that our method outperforms existing models with quadratic time complexity.” in the abstract, but it fails to demonstrate this in the results since lots of important models are not compared with on common graph benchmarks with smaller sizes. Even for scalable methods, many strong and relevant baselines are also not included such as:
- Efficient and Scalable Graph Generation through Iterative Local Expansion
- HIGEN: Hierarchical Graph Generative Networks
- Graph Generation with K2-Trees

**Questions:**

The paper claims to outperform quadratic-time methods while avoiding dropping too many edges during message passing. Beyond the fact that this is not sufficiently supported by experimental evidence. It is also unclear why the performance should be comparable, given the known expressivity gap between graph transformers and MPNNs. More clarification or theoretical justification would be helpful.

---

> ### Author Response · Authors · 2025-11-20
> **Response to Reviewer QS4y**
>
> **Clarification of Baselines, Comparisons, and Visualizations**
>
> We thank the reviewer for the thoughtful assessment of our work. As noted in our responses to other reviewers, we explored a broad range of models during development. We chose not to include certain baselines such as DiGress in cases where runs consistently resulted in out-of-memory failures; reporting incomplete or unstable results would not provide meaningful comparisons. Although we evaluated our model on smaller graphs and added a new section in the Appendix.
>
>
> Regarding flow-based generative models, such as Catflow, our focus in this paper is specifically on **diffusion-based large-scale graph generation. For this reason, we did not include flow-based methods as baselines.
>
> **Concerning models such as HiGen or K2-Trees** Although these approaches can generate graphs of up to roughly $1.5$K nodes (primarily point-cloud datasets), they do not support node or edge feature generation. In contrast, our datasets contain graphs with over $10{,}000$ nodes and require feature generation, making these models unsuitable for a fair or direct comparison—on top of the memory limitations we observed.
>
> We also agree that the table formatting in the original submission made comparisons less clear. In the revised version, we have restructured all tables for improved readability.
>
> The claim that our method outperforms models with quadratic time complexity refers to baselines such as GraphRNN ($O(N^2)$) and EDGE, which has a complexity of $O(T \max(M, K^2))$. In practice, methods like EDGE often rely on transformer-based modules with $O(N^2)$ complexity, and since $K^2$ frequently exceeds $M$, they effectively remain quadratic in computational cost.
>
> Regarding Figures A1--A3: the distinct visual themes were intentional. Each figure serves a different purpose (e.g., log-scale continuous axes vs.\ discrete axes), and the anomaly detection visualizations originate from a separate set of experiments. Nevertheless, we have **revised the edge-feature visualizations** for improved clarity and consistency.

---

> > ### Comment · Reviewer_QS4y · 2025-11-25
> >
> > Thank you for the response.
> > My key concerns remain unresolved, including the unclear motivation for different components of the work, missing strong baselines to support the claim in the abstract, the triviality of the directed extension, justification of the expressivity of the GNN being used, etc.
> > I therefore keep my original score.

---

> > > ### Author Response · Authors · 2025-11-26
> > > **Follow-up Response to Reviewer QS4y**
> > >
> > > Thank you for the follow-up. We would like to clarify the key points:
> > >
> > > As mentioned earlier, **the claim in the abstract** refers to baselines such as RNN-based models (e.g., GraphRNN) and methods like EDGE, whose complexity is $O(T \max(M, K^2))$. Since these approaches rely on transformer-style components with $O(N^2)$ cost—and typically $K^2 > M$—they remain effectively quadratic in practice. Our main contribution lies in improving the **scalability** of diffusion-based graph generative models, and our method is able to generate graphs with over 5k nodes and 80k edges, demonstrating this capability.
> > >
> > > **Motivation for the discriminator** The discriminator is introduced to enhance generation quality, following successful strategies in image diffusion models. In addition, it supports a meaningful secondary application in anomaly detection. We will make this motivation clearer in the paper.
> > >
> > > **Directed extension** The directed extension is motivated by practical considerations rather than by introducing a separate theoretical framework. We chose to model in-degrees explicitly because, in many real-world directed networks (e.g., citation, regulatory, and information-flow networks, and as illustrated by algorithms such as PageRank), in-degree carries stronger structural significance than out-degree. Our formulation follows this well-established perspective in complex network analysis, and the resulting extension is purposeful and meaningful rather than a trivial variant.
> > >
> > > We appreciate the reviewer’s perspective and will clarify these points in the final version.

---

### Official Review · Reviewer_m3WS · 2025-10-29

**Soundness:** 2
**Presentation:** 2
**Contribution:** 2
**Rating:** 4
**Confidence:** 3

**Summary:**

The paper proposes DGDGL, a model that utilizes discriminator-guided diffusion for generating large directed and Undirected Graphs. Its diffusion-based generator employs a message-passing graph neural network to synthesize node and edge features alongside structure, also lowering complexity from quadratic in the number of nodes to linear in the number of edges. The forward noising process involves iteratively removing edges from the input clean graph up to a theoretically argued minimum edge density threshold required for maintaining the effectiveness of message-passing. During sampling, the model iteratively denoises a corrupted graph into a clean one through iterative refinement and the removal of noisy edges. The discriminator's gradients guide the generator in denoising features and structure during sampling, and the authors provide theoretical support for the utility of the approach. Experimental results show that DDGL performs similarly to or better than various baselines on a set of large single-graph datasets spanning directed/undirected connectivity, as well as node/edge feature combinations.

**Strengths:**

- The proposed method enables the use of classic message passing graph neural network layers while achieving comparable performance to baselines with more complex network architectures.
- The concept of boosting sampling performance with a GAN-inspired discriminator is reasonable.
- The authors provide a theoretical discussion on specific properties, like the requirement for a minimum edge density to ensure GNNs operate effectively.

**Weaknesses:**

-The paper does not include any empirical results on the computation time of the proposed method relative to the baselines to supplement the claims of asymptotic complexity reduction.
-The evaluation could benefit from considering additional MMD measurements (e.g., spectre, orbit) common in other graph generation works (including baselines).
- Having at least one dataset composed of more than one graph could provide further insight into the model's generalization abilities, and a discussion on possible concerns about output diversity when integrating the discriminator in sampling.
- Certain notation elements and methodology details could benefit from some slight refinement and clarification (see questions).

**Questions:**

- Why does Table 1 not highlight any previous works that support node/edge classes, such as DiGress (mentioned elsewhere in the paper)? While DiGress does not focus on scalability, D-VAE also does not, yet it is present in the overview.
- In Table 1, the stated time complexity for DGDGL is O(|E|), which appears to be the per-time-step figure, rather than the end-to-end complexity of O(T|E|). For reference, in Algorithm 1, $f_\theta$, which has O(|E|) complexity per Appendix A3, is called T times for sampling. Conversely, for EDGE, the Table 1 complexity of O(T max(E, K^2)) appears to be end-to-end. Could the authors clarify the situation?
- Why is the value under “edge features” for Bitcoin-OTC set to 3 in Table 2, if the possible values are integers from -10 to 10?
- In Equation (A2), wouldn't the current notation imply that the node feature representation of each node gets concatenated to itself, which seems rather unhelpful?
- Given that the forward noising process is discrete, as it (mostly) removes edges from the clean graph, the fully-noised edges E_T should also form a discrete structure. However, in Algorithm 1 (and elsewhere in the manuscript), E_T gets sampled from a distribution N(...), which is generally associated with a continuous normal distribution. Is it correct that the distribution still represents a discrete prior, despite the notation?
- Similarly, if the forward process injects noise into a graph by mainly removing edges up to a minimum edge density $\rho^\text{min}_T$, and E_T gets initially sampled according to that density. However, the phrasing in line 253 and Equation (4) seems to imply that the sampling process also removes edges, which would further lower edge density. While the experiments show that the actual generation achieves average degrees comparable to, and often higher than, the clean graph, could the authors clarify the intent behind the aforementioned phrasing?

---

> ### Author Response · Authors · 2025-11-20
> **Response to Reviewer  m3WS**
>
> We thank the reviewer for the helpful and constructive feedback. To address your concern regarding the use of datasets containing multiple graphs and the associated notation, we have added new results and reformulated the noise operator in the revised version.
>
> **Why Table 1 Does Not Highlight Node/Edge Classes (e.g., DiGress)**
> Thank you for the question. Node and edge classes are indeed important in certain graph domains, such as molecular or heterogeneous graphs. However, our experiments focus on homogeneous graph settings, which is why this aspect is not emphasized in Table 1.
> Regarding DiGress: although it supports node and edge types, it does not scale to the graph sizes considered in our work. As discussed in our responses to other reviewers, DiGress has a computational complexity of $( O(TN^2) $), which makes generation on graphs larger than a few hundred nodes impractical. Its limitations include *convergent noise distribution* and *backbone complexity*. Further explanation is provided in our response to Reviewer 6NBZ.
> In practice, when running DiGress on larger datasets, we also observed out-of-memory failures. For these reasons, including DiGress would have resulted in incomplete or unstable baselines.
> **However**, in the revised version, we added new experiments on **smaller graphs** and evaluated our proposed model accordingly.
>
> Finally, D-VAE is included in Table 1 not for its scalability, but to illustrate the lack of *directed* scalable generative models. For undirected graphs, scalable methods such as EDGE exist (though they lack the capability for feature generation), but for directed graphs, this space remains largely unexplored.
>
> **Clarifying Complexity in Table 1 and the Role of $( T )$**
> We appreciate the reviewer’s careful reading. You are correct that each sampling step incurs $( O(|E|) $) cost, leading to an end-to-end figure of $( O(T|E|) $). In our method, incorporating $( T )$ does not change the asymptotic characterization because each step scales proportionally to $( |E| )$. That said, this distinction deserves to be stated clearly, and we will revise Table 1 and the accompanying discussion to avoid ambiguity.
>
> **Why Bitcoin-OTC Has 3 Edge Features in Table 2**
> Thank you for pointing this out. We distinguish between edge **labels** and edge **features**. The **labels** for Bitcoin-OTC and Bitcoin-Alpha indeed range from \(-10\) to \(+10\). However, the **features** correspond to temporal attributes (timestamp), decomposed into month, day, and hour. Thus, each edge has exactly 3 features, as reflected in Table 2 (lines 428--429 in the unrevised version).
>
> **Clarifying Equation (A2): No Self-Concatenation Intended**
> We agree with the reviewer that Equation (A2) was imprecisely expressed and unintentionally suggested “self-concatenation.” We apologize for the confusion. Our intention was to construct pairwise features for each node pair \((i, j)\) by concatenating their respective node features and feeding the result through an MLP to produce the pair representation. We have fully reformulated this section to clarify the intended operation.
>
> **Discrete Forward Process vs.\ Gaussian Notation for $( E_T $)**
> Thank you for this precise observation. The reviewer is correct: our initial manuscript used continuous Gaussian notation ($\mathcal{N}(\cdot) $) for a quantity that is in fact discrete. The forward process removes edges discretely, so $( E_T )$ should also be discrete.
> We have corrected this notational inconsistency throughout the revised manuscript.
>
> **Clarifying Edge Removal vs.\ Initialization Density in Equation (4)**
> We appreciate the reviewer’s question. In our formulation, edge removal constitutes the structural noise injection, and the model learns to identify which edges are realistic—that is, which edges are likely to belong to $( E_0 )$. This corresponds to an edge-pruning objective during training.
> During generation, however, $( E_T )$ is sampled as $( E_T \sim \mathrm{Bernoulli}(p^{\min}) )$.
> This initialization can yield a graph with a higher average degree than the original graph. For example, under an Erdős–Rényi model, the expected average degree is $( N \cdot p^{\min} )$, which may exceed that of the dataset. In our implementation, we set $( p^{\min} $) to a constant equal to $( N/E $), ensuring consistency across time steps. Since $( N < E $) for all our datasets, this choice naturally yields a denser initial graph.
> Thus, while the forward process removes edges when constructing $( E_t $), the sampling initialization $( E_T $) often starts with higher edge density. Our phrasing in the original manuscript was unclear, and we have now revised the text to better distinguish these two roles.

---

> > ### Comment · Reviewer_m3WS · 2025-11-25
> >
> > I would like to thank the authors for their answers to my questions and the extra results in Table A4.
> > In terms of experiments, I still believe that reporting empirical runtime, along with the complete set of MMD metrics across all tested scenarios, could enhance the paper’s strength.

---

> > > ### Author Response · Authors · 2025-11-28
> > > **Follow-up Response to Reviewer m3WS**
> > >
> > > **Thank you for the follow-up.**
> > >
> > > To address your concern, we have added additional results on *empirical sampling time* in the extended section. We evaluated generation time across different graph sizes and compared it with existing models. As shown in Figure~A6 in the revised version, our method maintains reasonable sampling efficiency even for large graphs—requiring under 110 seconds for graphs with more than 10k nodes—while models such as EDGE and GraphMaker require around 200 seconds under the same setting.

---

### Official Review · Reviewer_VH9p · 2025-10-30

**Soundness:** 2
**Presentation:** 2
**Contribution:** 2
**Rating:** 2
**Confidence:** 2

**Summary:**

The paper proposes a discriminator-guided diffusion framework for generating large graphs, both directed and undirected. The method introduces noise through edge deletion and feature perturbation, then uses a GNN-based denoising process to iteratively reconstruct graphs. A discriminator is added to the reverse process, providing gradient feedback meant to steer generation toward more “realistic” structures and attributes. The authors present theoretical conditions for maintaining graph connectivity under edge deletion and argue that discriminator guidance aligns the denoising with distributional objectives. Overall, the framework is positioned as a unified and scalable setup for graph generation, with an additional emphasis on anomaly detection as a downstream use case.

**Strengths:**

- The approach seems to yield good emprical results.

- Using the discriminator to perform anomaly detection is interesting. I think there are a couple of papers for anomaly detection using diffusion models, it could be nice to add a comparison with those methods.

**Weaknesses:**

The presentation of the method is rather unclear :

- First, you say that you approximate $p(G_{t-1} | G_t)$ using your reverse kernel. Yet, you claim this kernel is trained to reconstruct $G_0$ from $G_T$. No explanation is provided.

- Then, you refer $s_t$ and $\hat{e}_t$ as edge scores and noise prediction of the edge scores without further explanations. It's completely unclear what those variables are.

- It's very unclear what the denoising process consists in. What I gathered from the current manuscript is that the noising process removes edges, and denoising process iteratively "revise" edges from a randomly sampled graph. Does it allow to add edges or simply to remove some ?

- Proposition 3.1 is kind of trivial, as well as Corollary 3.2. It gives the impression that the authors included those statements for the sake of "more theory".

- In table 3, DGDGL is bolded for Degree on the Elliptic dataset, even though it is outperformed by EDGE and ARROW-Diff.

- Tables are sometimes hard to read. Since CPL and PLE are better when close to the ground truth, why not reporting a distance to the ground truth ? Also, consider including bolding in Table 5.

**Questions:**

- Noising in diffusion models is made under the assumption that different dimensions (e.g. tokens in text or pixels in images) are noised independently. Your operator breaks that assumption since it looks at density of the whole graph.

- See weaknesses concerning the denoising process.

- How does your approache reduce computational complexity in practice ? Does it treat graphs as fully connected

- Considering that you generate 32 samples, it would be nice to include error bars for your results.

- What is the practical relevance of chosen benchmarks ? What would be the real-world of generating such networks ?

---

> ### Author Response · Authors · 2025-11-20
> **Response to Reviewer VH9p**
>
> We thank the reviewer for the helpful and constructive feedback.
>
> **Independence Assumption in the Forward Noising Process**
>
>
> We thank the reviewer for the helpful comment. To avoid any ambiguity, we have improved the notation throughout the revised manuscript and added a new section that clearly explains the edge deletion operator $D_t$.
>
> Briefly, we note that the reviewer’s concern stems from a distinction between the change in overall density and the mechanism by which noise is applied. Our forward process remains fully factorized: (i) node and edge features are noised independently using the standard Variance-Preserving formulation, and (ii) each edge is removed through an independent Bernoulli trial at every step, with no dependence on neighboring edges or instantaneous graph density.
>
> The updated notation and the added explanation make this independence property explicit.
>
> **Clarifying Denoising Choices and Notation**
>
> We agree that some notation in the current version is unclear and will revise it for readability.
> Our design uses edge deletion as the sole structural noising mechanism to avoid the substantial computational overhead of adding edges during the forward pass.
>
> During sampling, we begin from a randomly structured graph (features sampled from ${N}(0, I)$ and structure sampled from a Bernoulli prior). The reverse process jointly denoises features and predicts the clean graph structure. A key observation is that feature denoising provides strong, high-fidelity signals that aid structural recovery. The optional discriminator guidance further refines realism, but the core generative capability comes from the coupled feature--structure denoising model.
>
> **Complexity: Sparse Operation vs.\ Fully Connected Graphs**
> *Do we treat graphs as fully connected?*
> No. Treating graphs as fully connected would incur $O(N^2)$ cost and is infeasible for large $N$. Our method never constructs or processes a full adjacency matrix.
>
> *How do we reduce computational complexity?*
> At each denoising step, the model operates on:
> $\text{(i) the existing edges in } E_t, \quad \text{and (ii) a sampled subset of non-edges}.$ This leads to complexity $O(|E_t|)$ rather than $O(N^2)$, enabling practical scalability.
>
>
> **Reporting Variance Across 32 Samples**
>
> We agree that variance measures are essential.
> In the revised manuscript, all tables now include standard deviations (computed across the 32 generated samples) alongside the mean, providing a clearer view of model stability.
>
> **Practical Relevance of the Benchmarks**
>
> The chosen benchmarks reflect a key real-world challenge: data scarcity and privacy. Many large-scale directed and undirected graphs—especially financial transaction networks—are proprietary, sensitive, or restricted due to regulatory concerns.
>
> Our goal is to provide a generative model capable of producing realistic synthetic graphs that can be safely shared and used for downstream research. The financial datasets used in the directed setting are representative of scenarios where access to real data is limited, yet synthetic alternatives can substantially accelerate research in areas such as financial crime analysis, fraud detection, and risk modeling.

---

> > ### Comment · Reviewer_VH9p · 2025-11-25
> >
> > Thanks for reporting error bars and the clarifications.
> >
> > That said, I still have strong concerns:
> >
> > - A sparse graph diffusion model, SparseDiff [1], already exists and is not mentioned. Such an important baseline should be included.
> > - The use of a discriminator to guide the generative process is not motivated.
> > - The theoretical treatment of the graph-related aspects is extremely superficial, relying on trivial observations.
> >
> > Therefore, I will keep my original score.
> >
> > [1] Sparse Training of Discrete Diffusion Models for Graph Generation, Qin et al, 2023

---

> > > ### Author Response · Authors · 2025-11-26
> > > **Follow-up Response to Reviewer VH9p**
> > >
> > > Thank you for the comments. We address the concerns as follows:
> > >
> > > **SparseDiff** While SparseDiff is related, its primary focus is on faster convergence rather than scalability. Even on datasets with up to 1045 nodes, the method samples query edges $E_q$ of size $\lceil \lambda N^2 \rceil$ with $\lambda = 0.1$ (best case), which is still quadratic in number of nodes. From our perspective, SparseDiff remains close to EDGE-style models with a message-passing backbone and does not provide the scalability benefits we target. Nonetheless, we will include it as a related work in the paper.
> > >
> > > **Motivation for the discriminator** Our use of a discriminator is motivated by improving generation quality, inspired by analogous techniques in image diffusion models. Moreover, the discriminator enables a meaningful secondary application in anomaly detection, which further justifies its inclusion. We will clarify this motivation more explicitly.
> > >
> > > **Theoretical discussion** Our goal was to formalize key observations about the interaction between the noising process and message-passing limitations (e.g., over-squashing), a topic not addressed in prior work such as EDGE or SparseDiff. We intentionally presented these points as a **discussion** rather than a full theoretical contribution for statements such as Corollary 3.2 and 3.3; had our aim been to develop a deeper theoretical framework, we would have included full proofs (which already exist in the GNN literature and are cited appropriately, though not in the context of generation) instead of explicitly labeling the section as ``Discussion of Corollary.'' Our intent was clarity rather than overstating the theoretical component, and we will further clarify this in the paper.

---

### Official Review · Reviewer_6NBZ · 2025-11-01

**Soundness:** 3
**Presentation:** 2
**Contribution:** 2
**Rating:** 4
**Confidence:** 3

**Summary:**

This paper proposes DGDGL, a diffusion-based generative model for creating large directed and undirected graphs with node and edge features. The key contribution is achieving linear O(|E|) complexity by injecting structural noise through edge deletion, combined with a GNN-based discriminator that provides gradient guidance during the reverse denoising process. The authors derive theoretical bounds on minimum edge density required for reconstruction. Experiments on citation networks and Bitcoin trust networks show competitive performance against baselines.

**Strengths:**

1. The proposed method utilizes a discriminator as guidance for graph generation. Both the results and ablation studies demonstrate that the proposed guidance term is effective.
2. The authors provide theoretical guarantees for edge density and GNN effectiveness.
3. The proposed method is lightweight. The authors employ only a 5-layer and 3-layer GNN architecture, yet achieve results comparable to existing baselines.

**Weaknesses:**

1. Although the proposed method demonstrates better computational complexity than baselines, the origin of the baseline complexity is unclear (i.e., whether it is cited from the original papers or computed by the authors). Additionally, no experimental results are provided to support the overhead comparison.
2. Discriminator guidance is not a novel technique for diffusion models, as it has been applied in image domain for a long time.
3. The mathematical notation could be improved, especially for the discrete components.

**Questions:**

1. How were the computational complexities of other baselines in Table 1 obtained? Were they taken from the original references or computed by the authors? If they were extracted from the references, this should be explicitly stated. If they were computed by the authors, the derivation process should be documented.
2. For EDGE, the number of generation steps T is included in the complexity analysis. However, why is the generation step count not included in the proposed method's complexity? Could you explain why the complexity is O(|E|) instead of O(T|E|)?
3. For previously mentioned methods like DiGress that are not included in the baselines, what are the specific reasons they cannot generate graphs larger than 300 nodes? Could these methods be scaled to large-scale graphs in the implementation presented in this paper?
4. Table 5 shows that increasing diffusion steps from 256 to 1024 leads to worse performance. This contradicts the typical behavior of diffusion models, where more steps generally lead to higher fidelity outputs. Does this indicate an instability in your reverse process, or does it suggest that the discriminator's guidance degrades over longer sampling chains?
5. Can the proposed method scale to larger graphs, such as those in OGB-LSC [1]?

[1] Hu, Weihua, et al. "Ogb-lsc: A large-scale challenge for machine learning on graphs." arXiv preprint arXiv:2103.09430 (2021).

---

> ### Author Response · Authors · 2025-11-20
> **Response to Reviewer 6NBZ**
>
> We thank the reviewer for the helpful and constructive feedback.
>
> **Clarifying Complexity Values (Questions 1 \& 2)**
> Thank you for the thoughtful questions. The complexity values reported in Table 1 are taken directly from the original papers or their cited references, and we will make this explicit in the revised version.
>
> Regarding the role of $T$: you are correct that the number of generation steps can be incorporated into the complexity analysis. In our method, including $T$ does not change the overall asymptotic complexity, as each step processes information proportional to $\lvert E \rvert$. We will add a more detailed explanation in the revision to avoid potential confusion.
>
> **Why DiGress Cannot Scale Beyond $\sim 300 $ Nodes**
> The primary reason DiGress does not scale to larger graphs is its computational complexity of $O(TN^2)$, which becomes prohibitive for large $N$. Two key limitations contribute:
>
> *1. Final noise distribution is dense.*
> DiGress uses a convergent noise distribution $G(\text{expected} \sim 0.5, N)$, which corresponds to a dense graph regardless of dataset sparsity. This forces the model to denoise dense large graphs, making large-scale sampling impractical. In contrast, our method uses a final noise distribution $G(p_t^{\min}, N)$ whose expected density matches that of the dataset, keeping noisy graphs computationally manageable. (See the EDGE paper for further discussion.)
>
> *2. Transformer-based backbone.*
> DiGress relies on a transformer GNN, incurring $O(N^2)$ cost. Replacing it with a standard GNN would improve scalability but significantly degrade performance. Our model instead uses node/edge features and discriminator feedback to adjust structure without relying on a quadratic-cost transformer encoder.
>
> In practice, we also encountered out-of-memory errors when running DiGress on larger datasets. Due to these issues, reporting incomplete or unstable results would have been misleading.
>
> **Effect of Increasing Diffusion Steps ($T$)**
> The results in Table 5 do not suggest instability. Increasing $T$ does not inherently worsen performance; rather, beyond a certain threshold (e.g., $T=256$), improvements become inconsistent. For example, *Citeseer* shows gains when increasing $T$ from 512 to 1024.
>
> Typically, diffusion models require additional training epochs to fully benefit from larger $T$. In Table~5, we fixed all other training settings to isolate the effect of varying $T$. Under this constraint, the marginal benefit of increasing the number of steps is limited, which explains why we do not observe the usual monotonic improvement.
>
> **Scalability to Extremely Large Graphs (e.g., OGB-LSC)**
> We agree that scaling to graphs of the size found in OGB-LSC (over 240M nodes) is fundamentally challenging. Even standard GNNs rely on node/edge sampling techniques based on similarity or dissimilarity (e.g., GraphSAGE [1]) for feasibility.
>
> In our setting, generation begins from a randomly initialized noisy graph, which provides no meaningful structure or neighborhoods from which to sample similar or dissimilar nodes. This makes scalable training intractable, and the model fails to converge.
>
> Therefore, the current version of our method does not scale to graphs of this size. Designing generative models for such massive graphs remains an important direction for future research.
>
>
>
> [1]. Hamilton, Will, Zhitao Ying, and Jure Leskovec. "Inductive representation learning on large graphs." Advances in neural information processing systems 30 (2017).

---

> > ### Comment · Reviewer_6NBZ · 2025-11-25
> >
> > I thank the authors for the answers to my questions and the clarifications. However, seeing the other reviews, I tend to agree with their concerns and plan to keep my score.

---

### Official Review · Reviewer_gwjt · 2025-11-01

**Soundness:** 2
**Presentation:** 3
**Contribution:** 3
**Rating:** 4
**Confidence:** 4

**Summary:**

The paper presents a diffusion model framework for graph generation consisting of

- a noise process constructed out of edge deletion (up to a sparsity threshold, uniformly adding edges back in once that threshold is hit) and noise injection

- co-training a discriminator that is used as a classifier guidance, distinguishing “realistic” from “nonrealistic” graphs and improving generation


The method is evaluated for graph generation on Citeseer,bitcoint-otc, elliptic,bitcoint-alpha,cora and pubmed datasets showing poimising results

Applies the idea of the above to graphs improves sampling, it’s good

**Strengths:**

clarity: the paper is well written and presents the ideas clearly

signfificance: the method shows overall promising results

originality: the paper is an (independent?) reinvention of [https://arxiv.org/abs/2211.17091](https://arxiv.org/abs/2211.17091) applied to graphs, which is a non-trivial step of novelty

quality: the paper is overall well written and evaluations make sense, with some caveats detailed below

**Weaknesses:**

- the paper is missing a related work [https://arxiv.org/abs/2211.17091](https://arxiv.org/abs/2211.17091) which should be added

- some of the evaluations metrics are very close, please train multiple models with more seeds and compute  CIs

- graphRNN is a weak baseline, should include GRAN and maybe  Bigg (dai et al)

- at least on the smaller datasets, should  to do isomorphism checks against the training dataset to guard against memorization

- figure 4 should do some form of rigorous evaluation (e.g. computing wasserstein distance between the distribution and the training distribution and comparing against  a gaussian/uniform prior, some form of statistical proximity ranking test). other figures that are making statistical points should be characterized like this as well

**Questions:**

see weaknesses, the main issue being the missing reference and more rigorous statistical evaluation

---

> ### Author Response · Authors · 2025-11-20
> **Response to Reviewer gwjt**
>
> We thank the reviewer for the helpful and constructive feedback.
>
> **Adding Missing Related Work**
> Thank you for pointing out this missing reference. We agree that the suggested work is relevant, and we have added it to the revised manuscript, along with a clarification of its connection to our approach.
>
> **Variance, Seeds, and Confidence Intervals**
> We appreciate the suggestion. We have conducted extensive experiments across a wide range of hyperparameters—learning rates, hidden dimensions, random seeds, and more. The paper reports the configurations most relevant for comparison.
>
> To address your concern, we can include standard deviations to better reflect variability in the reported metrics. For large graphs with over 3000 nodes, exhaustively tuning baselines becomes computationally expensive, but we followed recommended settings from the respective papers to ensure fair comparisons.
>
> **Choice of Baselines: GraphRNN vs.\ GRAN/BIGG**
> Thank you for raising this point. While GRAN and BIGG are more scalable (with $O(N)$ and $O((N+M)\log N)$ generation time), GRAN produced lower-quality samples than GraphRNN in our experiments. In addition, these models generally do not support node or edge feature generation. For these reasons, GraphRNN was used as the representative autoregressive baseline.
>
>
> **Isomorphism Checks to Prevent Memorization**
> We agree that checking for memorization is important. In addition to the edge overlap analysis for large graphs provided in the original version, we have now evaluated smaller datasets such as *Community* using isomorphism-based uniqueness measures, including the Weisfeiler–Lehman (WL) test. A new section has been added to the Appendix presenting results on both *Community* and *Ego*, where our model performs comparably to methods specifically designed for small-scale graph generation.
>
>
> **Rigorous Evaluation for Figure 4**
> Thank you for the comment. Our quantitative evaluation is captured through the metrics in the tables, which include both distribution-level and structural statistics. Figures such as Figure 3 and Figure 4 serve as qualitative visualizations to help readers interpret generated structures. For example, the MMD value for EdgeF in Table 3 numerically supports the patterns illustrated in Figure 4. We will clarify this connection more explicitly in the revision.

---

> > ### Comment · Reviewer_gwjt · 2025-11-25
> > **thank you for your rebuttal**
> >
> > 1. thank you
> > 2. I appreciate the difficulty of tuning things from a computational perspective, I think acknowledging this in the limitations sections is then enough (and puts the paper ahead of other work)
> > 3. I think adding node or edge feature generation should be trivial, but I think your rebuttal comment added as a footnote or appendix note would also be enough given you observed subpar performance
> > 4. nice, thank you very much, to clarify, this is comparing against the training set, and making an argument that with 30% of the training set not memorized, this should be evidence that there is no memorization?
> > 5. thank you

---

> > > ### Author Response · Authors · 2025-11-26
> > > **Follow-up Response to Reviewer gwjt**
> > >
> > > Thank you for the comments.
> > >
> > > We appreciate your suggestion about **node and edge feature** generation. Although generating meaningful node/edge features aligned with a given topology is not entirely straightforward, we will consider this and clarify it in the paper.
> > >
> > > It is worth noting that the training and test portions follow the same distribution and characteristics. To clarify, the key point is that the **uniqueness** metric on the test split and the generated graphs shows that our model does not simply duplicate training samples from the overall data distribution (train + test). If the model were memorizing (given the random selection of the test data), the uniqueness would be low, since the WL test reliably detects isomorphic or repeated structures.
> > >
> > > Taken together with the other quality metrics, this indicates that the model is not merely copying training graphs but is generating new ones.

---

### Author Response · Authors · 2025-12-02
**Summary of Reviewer Concerns and Our Responses**

We would like to thank the reviewers for their thoughtful and constructive feedback.
As highlighted in the manuscript, our work addresses the scalability challenge and feature generation in diffusion-based directed and undirected graph models. We enable the generation of large graphs (5k+ nodes) with node and edge features. Our main contributions are:

1) **Scalable Architecture with Guidance**: We use a lightweight, edge-linear GNN (for $O(|E|)$ complexity) and a sparse-preserving noise scheduler. Crucially, we introduce a *discriminator guidance* mechanism to correct the low expressiveness of this highly efficient backbone, enhancing both structural and feature quality without compromising scalability.
 2) **Theoretical Structure Preservation**: We analyze the effect of edge-deletion noise and derive a lower bound on edge density required to preserve connectivity. Our scheduler enforces this bound, guaranteeing reliable structural reconstruction.
 3)  **Validation and Efficiency**: Our model demonstrates strong generation performance on large, real-world directed and undirected graphs, validating its ability to handle complex features while maintaining linear computational complexity.

Below, we **summarize** the main concerns raised by the reviewers and our responses.

**Baselines**
We appreciate the concerns regarding baseline selection, but argue that within our setting the selection is appropriate. Our choices were guided by fairness, practicality, and alignment with the large-scale setting of this work.

1) For comparability, we excluded models that cannot generate graphs exceeding 3k nodes, as these methods either fail or run out of memory in this regime.
2) We selected representative methods from major model families. For instance, GraphRNN serves as the autoregressive baseline because it consistently performs strongest within its class. Models lacking feature generation or failing to scale beyond 3k nodes were omitted. (Notably, some prior works refer to graphs with only 1k--2k nodes as “large,” e.g., HiGen and K2-Trees, which is outside the scale considered here.)
3) Many existing methods do not report results on *directed* graphs, making direct comparison infeasible.

Overall, our baseline selection prioritizes fairness, reproducibility, and suitability for the large-scale graph generation setting addressed in this paper.

**Results on Smaller Graphs**
Reviewers also requested results on smaller graphs to compare against more existing models and to evaluate metrics such as **uniqueness** and **novelty**. In the rebuttal, we conducted additional experiments on smaller datasets including *Ego* and *Community*, and reported uniqueness and novelty metrics to assess sample diversity. The results show that our model performs competitively with methods like DiGress, which are specifically designed for smaller graphs, while methods like DiGress do not scale to the datasets we target with our method.

**Sampling Time in Practice**
Another concern was practical sampling efficiency, so we reported runtimes across a wide range of graph sizes. Our model can generate graphs with over 10k nodes in under 110 seconds—performance that is comparable to, and in some cases surpasses, existing baselines.

**Clarification of the Discriminator Motivation**
For scalability, the reverse process is intentionally kept linear in the number of edges. The discriminator is introduced to support this lightweight architecture by providing gradient-based guidance that improves learning stability and generation quality. Our ablation study, which varies the strength of guidance, clearly demonstrates its positive effect. We agree that this motivation was not sufficiently emphasized in the original submission; the revised version now clarifies this rationale and cites related work from other domains (e.g., image generation) where similar guidance has proven effective.

**Directed Graph Extension**
Our directed-graph formulation is grounded in explicitly modeling *in-degrees*, motivated by practical considerations rather than by developing an entirely separate theory. In many real-world directed networks—such as citation graphs, regulatory systems, and information-flow structures—in-degree carries more structural significance than out-degree. This perspective is well established in complex network analysis (e.g., the PageRank algorithm). Our extension leverages this insight, making it a purposeful and meaningful modeling choice rather than a trivial modification. Moreover, as noted in the Limitations section, the absence of out-degree modeling remains a limitation and is an avenue for future work.


We believe that the new experiments, quantitative results, and clarifications comprehensively address all major reviewer concerns and demonstrate the novelty, performance, and practical scalability of our proposed model. We look forward to your decision.

---

### Meta-Review · Area_Chair_Zm3W · 2026-01-07

**Summary:**

This paper proposes DGDGL, a diffusion-based generative model for creating large directed and undirected graphs with node and edge features which unifies structure and feature generation for both nodes and edges within a single framework. The authors also provide theoretical guarantees for structure preservation. Experiments on six real-world datasets display competitive performances.

The reviewers of this paper provided detailed feedback. The main concerns raised are as follows:
1. Novelty limitation of discriminator guidance.
2. The mathematical notation and table can be improved.
3. More experiments and state-of-the-art baselines are required.

**Reviewer Concerns:**

All reviewers' comments are partially addressed during the rebuttal stage.

**Reviewer Scores:**

They may not re-evaluate their scores.

---

### Decision · Program_Chairs · 2026-01-26

Reject